# Domain Generalisation via Domain Adaptation: An Adversarial Fourier Amplitude Approach

**Minyoung Kim**[1], **Da Li**[1] **& Timothy M. Hospedales**[1,2]
[1]Samsung AI Center Cambridge, UK                    [2]University of Edinburgh, UK
{mikim21,dali.academic}@gmail.com         t.hospedales@ed.ac.uk

## Abstract

We tackle the domain generalisation (DG) problem by posing it as a domain adaptation (DA) task where we adversarially synthesise the worst-case 'target' domain and adapt a model to that worst-case domain, thereby improving the model's robustness. To synthesise data that is challenging yet semantics-preserving, we generate Fourier amplitude images and combine them with source domain phase images, exploiting the widely believed conjecture from signal processing that amplitude spectra mainly determines image style, while phase data mainly captures image semantics. To synthesise a worst-case domain for adaptation, we train the classifier and the amplitude generator adversarially. Specifically, we exploit the maximum classifier discrepancy (MCD) principle from DA that relates the target domain performance to the discrepancy of classifiers in the model hypothesis space. By Bayesian hypothesis modeling, we express the model hypothesis space effectively as a posterior distribution over classifiers given the source domains, making adversarial MCD minimisation feasible. On the DomainBed benchmark including the large-scale DomainNet dataset, the proposed approach yields significantly improved domain generalisation performance over the state-of-the-art.

## 1    Introduction

Contemporary machine learning models perform well when training and testing data are identically distributed. However, in practice it is often impossible to obtain an unbiased sample of real-world data for training, and therefore distribution-shift inevitably exists between training and deployment. Performance can degrade dramatically under such domain shift (Koh et al., 2021), and this is often the cause of poor performance of real-world deployments (Geirhos et al., 2020). This important issue has motivated a large amount of research into the topic of domain generalisation (DG) (Zhou et al., 2021a), which addresses training models with increased robustness to distribution shift. These DG approaches span a diverse set of strategies including architectural innovations (Chattopadhyay et al., 2020), novel regularisation (Balaji et al., 2018), alignment (Sun & Saenko, 2016) and learning (Li et al., 2019) objectives, and data augmentation (Zhou et al., 2021b) to make available training data more representative of potential testing data. However, the problem remains essentially unsolved, especially as measured by recent carefully designed benchmarks (Gulrajani & Lopez-Paz, 2021).

Our approach is related to existing lines of work on data-augmentation solutions to DG (Zhou et al., 2021b; Shankar et al., 2018), which synthesise more data for model training; and alignment-based approaches to Domain Adaptation (Sun & Saenko, 2016; Saito et al., 2018) that adapt a source model to an unlabeled target set – but cannot address the DG problem where the target set is unavailable. We improve on both by providing a unified framework for stronger data synthesis and domain alignment.

Our framework combines two key innovations: A Bayesian approach to maximum classifier discrepancy, and a Fourier analysis approach to data augmentation. We start from the perspective of maximum classifier discrepancy (MCD) from domain adaptation (Ben-David et al., 2007; 2010; Saito et al., 2018). This bounds the target-domain error as a function of discrepancy between multiple source-domain classifiers. It is not obvious how to apply MCD to the DG problem where we have no access to target-domain data. A key insight is that MCD provides a principled objective that we can maximise in order to *synthesise* a worst-case target domain, and also minimise in order to train a model that is adapted to that worst-case domain. Specifically, we take a Bayesian approach that learns

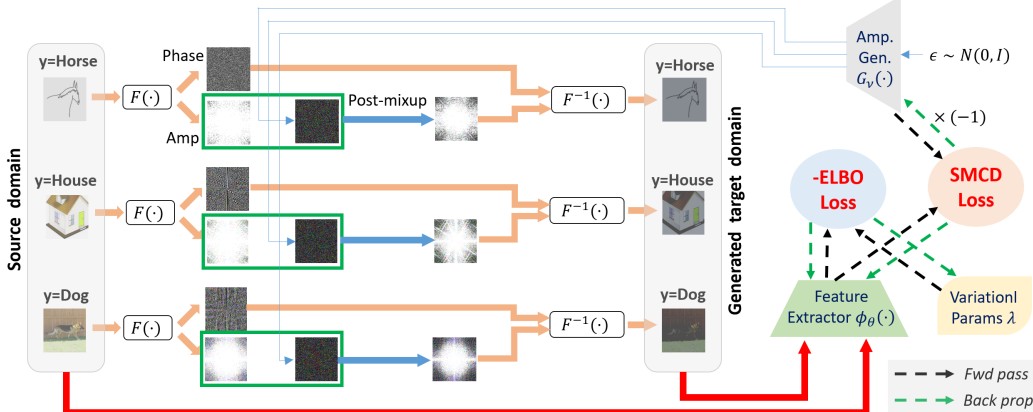

Figure 1: Overall training flow of the proposed approach (AGFA). We generate target-domain data by synthesizing Fourier amplitude images trained adversarially. See main text in Sec. 3 for details.

a distribution over source-domain classifiers, with which we can compute MCD. This simplifies the model by eliminating the need for adversarial classifier training in previous applications of MCD (Saito et al., 2018), which leaves us free to adversarially train the worst-case target domain. To enable challenging worst-case augmentations to be generated without the risk of altering image semantics, our augmentation strategy operates in the Fourier amplitude domain. It synthesises amplitude images, which can be combined with phase images from source-domain data to produce images that are substantially different in style (amplitude), while retaining the original semantics (phase). Our overall strategy termed Adversarial Generation of Fourier Amplitude (AGFA) is illustrated in Fig. 1.

In summary, we make the following main contributions: (1) We provide a novel and principled perspective on DG by drawing upon the MCD principle from DA. (2) We provide AGFA, an effective algorithm for DG based on variational Bayesian learning of the classifier and Fourier-based synthesis of the worst-case domain for robust learning. (3) Our empirical results show clear improvement on previous state-of-the-arts on the rigorous DomainBed benchmark.

## 2  PROBLEM SETUP AND BACKGROUND

We follow the standard setup for the Domain Generalisation (DG) problem. As training data, we are given labeled data $S = \{(x, y) | (x, y) \sim \mathcal{D}_i, i = 1, \dots, N\}$ where $x \in \mathcal{X}$ and $y \in \mathcal{Y} = \{1, \dots, C\}$. Although the source domain $S$ consists of different domains $\{\mathcal{D}_i\}_{i=1}^N$ with domain labels available, we simply take their union without using the originating domain labels. This is because in practice the number of domains ($N$) is typically small, and it is rarely possible to estimate a meaningful population distribution for empirical $S$ from a few different domains. What distinguishes DG from the closely-related (unsupervised) Domain Adaptation (DA), is that the target domain ($T$) on which model's prediction performance is measured is *unknown* for DG, whereas in DA the input data $x$ from the target domain are revealed (without class labels $y$). Below we briefly summarise the MCD principle and Ben-David's theorem, one of the key theorems in DA, as we exploit them to tackle DG.

**Ben-David's theorem and MCD principle in DA.**  In unsupervised DA, Ben-David's theorem (Ben-David et al., 2010; 2007) provides an upper bound for the target-domain generalisation error of a model (hypothesis). We focus on the tighter bound version, which states that for any classifier $h$ in the hypothesis space $\mathcal{H} = \{h | h : \mathcal{X} \to \mathcal{Y}\}$, the following holds (without the sampling error term):

$$e_T(h) \le e_S(h) + \sup_{h, h' \in \mathcal{H}} \left| d_S(h, h') - d_T(h, h') \right| + e^*(\mathcal{H}; S, T), \qquad (1)$$

where $e_S(h) := \mathbb{E}_{(x,y) \sim S}[\mathbb{I}(h(x) \ne y)]$ is the error rate of $h(\cdot)$ on the source domain $S$, $d_S(h, h') := \mathbb{E}_{x \sim S}[\mathbb{I}(h(x) \ne h'(x))]$ denotes the discrepancy between two classifiers $h$ and $h'$ on $S$ (similarly for $e_T(h)$ and $d_T(h, h')$), and $e^*(\mathcal{H}; S, T) := \min_{h \in \mathcal{H}} e_S(h) + e_T(h)$. Thus we can provably reduce the target domain generalisation error by simultaneously minimizing the three terms in the upper bound[1], namely source-domain error $e_S(h)$, classifier discrepancy, and minimal source-target error.

Previous approaches (Saito et al., 2018; Kim et al., 2019) aim to minimise the upper bound, and one reasonable strategy is to *constrain* the hypothesis space $\mathcal{H}$ in such a way that it contains only those

---

[1]Some recent work such as (Vedantam et al., 2021), however, empirically studied potential risk of looseness of the bound in certain scenarios.

$h$'s with small $e_S(h)$. Within this source-confined hypothesis space (denoted by $\mathcal{H}_{|S}$), the terms $e_S(h)$ and $d_S(h, h')$ in the bound are expected to be close to 0 for all $h, h' \in \mathcal{H}_{|S}$, and the bound of (1) effectively reduces to what is called the *Maximum Classifier Discrepancy* (MCD) loss,

$$\text{MCD}(\mathcal{H}_{|S}; T) := \sup_{h, h' \in \mathcal{H}_{|S}} |d_T(h.h')| = \sup_{h, h' \in \mathcal{H}_{|S}} \mathbb{E}_{x \sim T} \big[ \mathbb{I}(h(x) \neq h'(x)) \big]. \tag{2}$$

This suggests the **MCD learning principle**: we need to minimise both the error on $S$ (so as to form the source-confined hypothesis space $\mathcal{H}_{|S}$) and the MCD loss on $T$. Note however that the last term $e^*$ is not considered in (Saito et al., 2018; Kim et al., 2019) mainly due to the difficulty of estimating the target domain error. We will incorporate $e^*$ in our DG algorithm as described in the next section.

We conclude the section by briefly reviewing how the MCD learning principle was exploited in previous works. In (Saito et al., 2018) they explicitly introduce two classifier networks $h(x) = g(\phi(x))$ and $h'(x) = g'(\phi(x))$, where the classification heads $g, g'$ and the feature extractor $\phi$ are *cooperatively* updated to minimise the error on $S$ (thus implicitly obtaining $\mathcal{H}_{|S}$), they are updated *adversarially* to maximise (minimise) the MCD loss on $T$ with respect to $g$ and $g'$ ($\phi$, respectively). In (Kim et al., 2019), they build a Gaussian process (GP) classifier on the feature space $\phi(x)$, in which $\mathcal{H}_{|S}$ is attained by GP posterior inference. Minimisation of the MCD term is then accomplished by the maximum margin learning which essentially enforces minimal overlap between the two largest posterior modes. Note that (Saito et al., 2018)'s strategy requires adversarial optimisation, and hence it is less suitable for our DG algorithm which will require adversarial generator learning: Having two adversarial learning components would make the training difficult since we need to find two nested equilibrium (saddle) points. We instead adopt the Bayesian hypothesis modeling approach of (Kim et al., 2019). In the next section, we describe our approach in greater detail.

## 3 Adversarial Generation of Fourier Amplitude (AGFA)

**Defining and optimising a hypothesis space.**  Our DG approach aims to minimise the MCD loss, $\text{MCD}(\mathcal{H}_{|S}; T)$ defined in (2). The first challenge is that the target domain data $T$ is not available in DG. Before we address it, we clarify the optimisation problem (i.e., what is the MCD loss optimised for?) and how the hypothesis spaces ($\mathcal{H}$ and $\mathcal{H}_{|S}$) are represented. The MCD loss is a function of *hypothesis space* $\mathcal{H}$ (or $\mathcal{H}_{|S}$), not a function of individual classifier $h$ in it. Hence, minimising the MCD loss amounts to choosing the best hypothesis space $\mathcal{H}$. To this end, we need to parametrise the hypothesis space (so as to frame it as a continuous optimisation), and our choice is the *Bayesian linear classifier with deterministic feature extractor*.

We consider the conventional neural-network feed-forward classifier modeling: we have the feature extractor network $\phi_\theta(x) \in \mathbb{R}^d$ (with the weight parameters $\theta$) followed by the linear classification head $W = [w_1, \ldots, w_C]$ ($C$-way classification, each $w_j \in \mathbb{R}^d$), where the class prediction is done by the softmax likelihood:

$$P(y = j | x, \theta, W) \propto e^{w_j^\top \phi_\theta(x)}, \quad j = 1, \ldots, C. \tag{3}$$

So each configuration $(\theta, W)$ specifies a particular classifier $h$. To parametrise the hypothesis space $\mathcal{H}$ ($\ni h$), ideally we can consider *a parametric family of distributions* over $(\theta, W)$. Each distribution $P_\beta(\theta, W)$ specified by the parameter $\beta$ corresponds to a particular hypothesis space $\mathcal{H}$, and each sample $(\theta, W) \sim P_\beta(\theta, W)$ corresponds to a particular classifier $h \in \mathcal{H}$. Although this is conceptually simple, to have a tractable model in practice, we define $\theta$ to be *deterministic* parameters and only $W$ to be stochastic. A reasonable choice for $P(W)$, without any prior knowledge, is the standard Gaussian, $P(W) = \prod_{j=1}^{C} \mathcal{N}(w_j; 0, I)$.

Now, we can represent a hypothesis space as $\mathcal{H} = \{P(y|x, \theta, W) \mid W \sim P(W)\}$. Thus $\mathcal{H}$ is parametrised by $\theta$, and with $\theta$ fixed ($\mathcal{H}$ fixed), each sample $W$ from $P(W)$ instantiates a classifier $h \in \mathcal{H}$. The main benefit of this Bayesian hypothesis space modeling is that we can induce the *source-confined hypothesis space* $\mathcal{H}_{|S}$ (i.e., the set of classifiers that perform well on the source domain) in a principled manner by the posterior,

$$P(W|S, \theta) \propto P(W) \cdot \prod_{(x,y) \sim S} P(y|x, \theta, W). \tag{4}$$

The posterior places most of its probability density on those samples (classifiers) $W$ that attain high likelihood scores on $S$ (under given $\theta$) while being smooth due to the prior. To ensure that the source

domain $S$ is indeed explained well by the model, we further impose high data likelihood on $S$ as constraints for $\theta$,

$$\theta \in \Theta_S \text{ where } \Theta_S := \{\theta \mid \log P(S|\theta) \geq L_{th}\}, \tag{5}$$

where $L_{th}$ is the (constant) threshold that guarantees sufficient fidelity of the model to explaining $S$. Then it is reasonable to represent $\mathcal{H}_{|S}$ by the support of $P(W|S, \theta)$ for $\theta \in \Theta_S$, postulating that $\mathcal{H}_{|S}$ exclusively contains smooth classifiers $h$ that perform well on $S$. Formally, the source-confined hypothesis space can be parametrised as:

$$\mathcal{H}_{|S}(\theta) = \{P(y|x, \theta, W) \mid W \sim P(W|S, \theta)\} \text{ for } \theta \in \Theta_S, \tag{6}$$

where we use the notation $\mathcal{H}_{|S}(\theta)$ to emphasise its dependency on $\theta$. Intuitively, the hypothesis space $\mathcal{H}_{|S}$ is identified by choosing the feature space (i.e., choosing $\theta \in \Theta_S$), and individual classifiers $h \in \mathcal{H}_{|S}$ are realised by the Bayesian posterior samples $W \sim P(W|S, \theta)$ (inferred on the chosen feature space). Since the posterior $P(W|S, \theta)$ in (6) and the marginal likelihood $\log P(S|\theta)$ in (5) do not admit closed forms in general, we adopt the variational inference technique to approximate them. We defer the detailed derivations (Sec. 3.1) for now, and return to the MCD minimisation problem since we have defined the hypothesis space representation.

**Optimising a worst-case target domain.**   For the DG problem, we cannot directly apply the MCD learning principle since the target domain $T$ is unknown during the training stage. Our key idea is to consider the worst-case scenario where the target domain $T$ maximises the MCD loss. This naturally forms minimax-type optimisation,

$$\min_{\theta \in \Theta_S} \max_{T} \text{ MCD}(\mathcal{H}_{|S}(\theta); T). \tag{7}$$

To solve the saddle-point optimisation (7), we adopt the adversarial learning strategy with a generator network (Goodfellow et al., 2014). The generator for $T$ has to synthesise samples $x$ of $T$ that need to satisfy three conditions: (**C1**) The generated samples maximally baffle the classifiers in $\mathcal{H}_{|S}$ to have least consensus in prediction (for inner maximisation); (**C2**) $T$ still retains the same semantic class information as the source domain $S$ (for the definition of DG); and (**C3**) The generated samples in $T$ need to be distinguishable along their classes[2].

**Paramaterising domains.**   To meet these conditions, we generate target domain images using Fourier frequency spectra. We specifically build a generator network that synthesises *amplitude* images in the Fourier frequency domain. The synthesised amplitude images are then combined with the *phase* images sampled from the source domain $S$ to construct new samples $x \in T$ by inverse Fourier transform. This is motivated by signal processing where it is widely believed that the frequency phase spectra capture the semantic information of signals, while the amplitudes take charge of non-semantic (e.g., style) aspects of the signals (Oppenheim & Lim, 1981). Denoting the amplitude generator network as $G_\nu(\epsilon)$ with parameters $\nu$ and random noise input $\epsilon \sim \mathcal{N}(0, I)$, our target sampler $(x, y) \sim T$ are generated as follows:

1. $(x_S, y_S) \sim S$  (Sample an image and its class label from $S$)
2. $A_S \angle P_S = \mathcal{F}(x_S)$  (Fourier transform to have amplitude and phase for $x_S$)
3. $A = G_\nu(\epsilon), \epsilon \sim \mathcal{N}(0, I)$  (Generate an amplitude image from $G$)
4. $x = \mathcal{F}^{-1}(A \angle P_S), y = y_S$  (Construct target data with the synthesised $A$)

Here, $\mathcal{F}(\cdot)$ is the 2D Fourier transform, $F(u, v) = \mathcal{F}(x) = \iint x(h, w)e^{-i(hu+wv)}dhdw$, and $A \angle P$ stands for the polar representation of the Fourier frequency responses (complex numbers) for the amplitude image $A$ and the phase image $P$. That is, $A \angle P = A \cdot e^{i \cdot P} = A \cdot (\cos P + i \sin P)$ with $i = \sqrt{-1}$, where all operations are element/pixel-wise. Note that we set $y = y_S$ in step 4 since the original phase (semantic) information $P_S$ is retained in the synthesised $x$.

**Algorithm summary.**   Finally the worst-case target MCD learning can be solved by adversarial learning, which can be implemented as an alternating optimisation:

$$(\text{Fix } \nu) \quad \min_{\theta \in \Theta_S} \text{ MCD}(\mathcal{H}_{|S}(\theta); T(\nu)) \tag{8}$$

$$(\text{Fix } \theta) \quad \max_{\nu} \text{ MCD}(\mathcal{H}_{|S}(\theta); T(\nu)) \tag{9}$$

---

[2]This condition naturally originates from the solvability of the DG problem.

We used $T(\nu)$ to emphasise functional dependency of target images on the generator parameters $\nu$. Note that although the MCD loss in DA can be computed *without* the target domain labels (recall the definition (2)), in our DG case the class labels for the generated target data are available, as induced from the phase $P_S$ (i.e., $y = y_S$ in step 4). Thus we can modify the MCD loss by incorporating the target class labels. In the following we provide concrete derivations using the variational posterior inference, and propose a modified MCD loss that takes into account the induced target class labels.

## 3.1 CONCRETE DERIVATIONS USING VARIATIONAL INFERENCE

**Source-confined hypothesis space by variational inference.** The posterior $P(W|S, \theta)$ does not admit a closed form, and we approximate $P(W|S, \theta)$ by the Gaussian variational density,

$$Q_\lambda(W) = \prod_{j=1}^{C} \mathcal{N}(w_j; m_j, V_j), \tag{10}$$

where $\lambda := \{m_j, V_j\}_{j=1}^{C}$ constitutes the variational parameters. To enforce $Q_\lambda(W) \approx P(W|S, \theta)$, we optimise the evidence lower bound (ELBO),

$$\text{ELBO}(\lambda, \theta; S) := \sum_{(x,y) \sim S} \mathbb{E}_{Q_\lambda(W)}\big[\log P(y|x, W, \theta)\big] - \text{KL}\big(Q_\lambda(W)\|P(W)\big), \tag{11}$$

which is the lower bound of the marginal data likelihood $\log P(S|\theta)$ (Appendix A.3 for derivations). Hence maximising $\text{ELBO}(\lambda, \theta; S)$ with respect to $\lambda$ tightens the posterior approximation $Q_\lambda(W) \approx P(W|S, \theta)$, while maximising it with respect to $\theta$ leads to high data likelihood $\log P(S|\theta)$. The latter has the very effect of imposing the constraints $\theta \in \Theta_S$ in (8) since one can transform constrained optimisation into a regularised (Lagrangian) form equivalently (Boyd & Vandenberghe, 2004).

**Optimising the MCD loss.** The next thing is to minimise the MCD loss, $\text{MCD}(\mathcal{H}_{|S}(\theta); T)$ with the current target domain $T$ generated by the generator network $G_\nu$. That is, solving (8). We follow the maximum margin learning strategy from (Kim et al., 2019), where the idea is to enforce the prediction consistency for different classifiers (i.e., posterior samples) $W \sim Q_\lambda(W)$ on $x \sim T$ by separating the highest class score from the second highest by large margin. To understand the idea, let $j^*$ be the model's predicted class label for $x \sim T$, or equivalently let $j^*$ have the highest class score $j^* = \arg\max_j w_j^\top \phi(x)$ as per (3). (We drop the subscript in $\phi_\theta(x)$ for simplicity in notation.) We let $j^\dagger$ be the second most probable class, i.e., $j^\dagger = \arg\max_{j \neq j^*} w_j^\top \phi(x)$. Our model's class prediction would change if $w_{j^*}^\top \phi(x) < w_{j^\dagger}^\top \phi(x)$ for some $W \sim Q_\lambda(W)$, which leads to *discrepancy* of classifiers. To avoid such overtaking, we need to ensure that the (plausible) *minimal* value of $w_{j^*}^\top \phi(x)$ is greater than the (plausible) *maximal* value of $w_{j^\dagger}^\top \phi(x)$. Since the score (logit) $f_j(x) := w_j^\top \phi(x)$ is Gaussian under $Q_\lambda(W)$, namely

$$f_j(x) \sim \mathcal{N}(\mu_j(x), \sigma_j(x)^2) \text{ where } \mu_j(x) = m_j^\top \phi(x), \ \sigma_j^2(x) = \phi(x)^\top V_j \phi(x), \tag{12}$$

the prediction consistency is achieved by enforcing: $\mu_{j^*}(x) - \alpha\sigma_{j^*}(x) > \mu_{j^\dagger}(x) + \alpha\sigma_{j^\dagger}(x)$, where we can choose $\alpha = 1.96$ for 2.5% rare one-sided chance. By introducing slack variables $\xi(x) \geq 0$,

$$\mu_{j^*}(x) - \alpha\sigma_{j^*}(x) \geq 1 + \max_{j \neq j^*}\big(\mu_j(x) + \alpha\sigma_j(x)\big) - \xi(x). \tag{13}$$

Satisfying the constraints amounts to fulfilling the desideratum of MCD minimisation, essentially imposing prediction consistency of classifiers. Note that we add the constant 1 in the right hand side of (13) for the normalisation purpose to prevent the scale of $\mu$ and $\sigma$ from being arbitrary small. The constraints in (13) can be translated into the following MCD loss (as a function of $\theta$):

$$\text{MCD}(\theta; T) := \mathbb{E}_{x \sim T}\Big(1 + \mathcal{T}^2\big(\mu_j(x) + \alpha\sigma_j(x)\big) - \mathcal{T}^1\big(\mu_j(x) - \alpha\sigma_j(x)\big)\Big)_+ \tag{14}$$

where $\mathcal{T}^k$ is the operator that selects the top-$k$ element, and $(a)_+ = \max(0, a)$.

**Modified MCD loss.** The above MCD loss does not utilise the target domain class labels $y = y_S$ that are induced from the phase information $P_S$ (Recall the target domain data generation steps $1 \sim 4$ above). To incorporate the supervised data $\{(x, y)\} \in T$ in the generated target domain, we modify

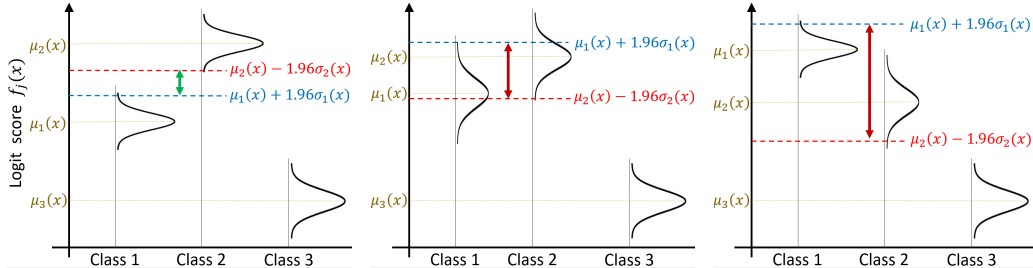

Figure 2: Illustration of the SMCD loss on three different hypothesis spaces $\mathcal{H}_{|S}$ shown in three panels. For $C = 3$-way classification case, each panel shows the class logit scores (Gaussian random) $f_j(x) \sim \mathcal{N}(\mu_j(x), \sigma_j(x)^2)$ for $j = 1, 2, 3$, at some input $x \in T$. We assume that the true (induced) class label $y = 2$. (**Left**) Since the mean logit for class 2, $\mu_2(x)$ is the maximum among others, the prediction is *marginally* correct (from softmax). Beyond that, the logit of the worst plausible hypothesis for class 2, $\mu_2(x) - 1.96\sigma_2(x)$ is greater than that of the runner-up class 1, $\mu_1(x) + 1.96\sigma_1(x)$ by some positive margin (green arrow), meaning there is little chance of prediction overtaking (so, consistent); equivalently, the SMCD loss is small. (**Middle**) Prediction is marginally correct, but prediction overtaking is plausible, indicated by the negative margin (red arrow); the SMCD loss is large. (**Right**) Incorrect marginal prediction (to class 1) with more severe negative margin (red arrow); the SMCD loss is even larger.

the MCD loss as follows: First, instead of separating the margin between the two largest logit scores as in the MCD, we maximise the margin between the logit for the given class $y$ and the largest logit among the classes other than $y$. That is, we replace the constraints (13) with the following:

$$\mu_y(x) - \alpha\sigma_y(x) \geq 1 + \max_{j \neq y} \left( \mu_j(x) + \alpha\sigma_j(x) \right) - \xi(x), \tag{15}$$

where $y$ is the class label (induced from the phase information) for the generated instance $x$. See Fig. 2 for illustration of the idea. Consequently, our new MCD loss (coined *supervised* MCD or SMCD for short) is defined as follows:

$$\text{SMCD}(\theta; T) := \mathbb{E}_{(x,y) \sim T} \left( 1 + \max_{j \neq y} \left( \mu_j(x) + \alpha\sigma_j(x) \right) - \left( \mu_y(x) - \alpha\sigma_y(x) \right) \right)_+ . \tag{16}$$

Here the variational parameters $\lambda$ is treated as constant since the only role of $\lambda$ is to maximise the ELBO. It should be noted that (16) essentially aims at maximising the logit for the given class $y$ (the last term), or equivalently, classification error minimisation on $T$, and at the same time minimising the logit for the runner-up class (the middle max term). Surprisingly, the former amounts to minimising the minimal source-target error term $e^*(\mathcal{H}; S, T)$ in the generalisation bound (1), which we have left out so far. That is, $e^*(\mathcal{H}; S, T) = \min_{h \in \mathcal{H}} e_S(h) + e_T(h) \approx \min_{h \in \mathcal{H}_{|S}} e_T(h)$, and the last term of the SMCD loss leads to $\theta$ that makes $e_T(h)$ small for all $h \in \mathcal{H}_{|S}(\theta)$. Moreover, minimising the logit for the runner-up class (the middle max term of the SMCD) has the effect of margin maximisation.

**Algorithm summary.** Our AGFA algorithm can be understood as *MCD-based DA with adversarial amplitude generated target domain*. It entails the following alternating optimisation ($\eta > 0$ is the trade-off hyperparameter for SMCD):

1. $\min_{\lambda, \theta} -\text{ELBO}(\lambda, \theta; S) + \eta\text{SMCD}(\theta; T)$      (model learning + VI; $\nu$ fixed)
2. $\max_{\nu} \text{SMCD}(\theta; T)$            (adversarial generator learning; $\theta$, $\lambda$ fixed)

Our algorithm is summarised in Alg. 1 (in Appendix) and illustrated schematically in Fig. 1. At test time, we can apply the classifier (3) with the learned $\theta$ and any sample $W \sim Q_\lambda(W)$ to target domain inputs to predict class labels. In our experiments, we take the posterior means $w_j = m_j$ instead of sampling from $Q_\lambda(W)$.

## 3.2 FURTHER CONSIDERATIONS

**Post-synthesis mixup of generated amplitude images.** In our adversarial learning, the amplitude generator network $G_\nu$ synthesises target domain image samples that have highly challenging amplitude spectra to the current model. Although we retain the phase information from source domains, unconstrained amplitude images can potentially alter the semantic content destructively (e.g., a constant zero amplitude image would zero out the image content), rendering it impossible to classify. To this end, instead of using the generator's output $A = G_\nu(\epsilon)$ directly, we combine it with

the source domain amplitude image corresponding to the phase image by simple mixup. That is, by letting $A_S$ be the amplitude spectra corresponding to the phase $P_S$, we alter $A$ as:

$$A \leftarrow \lambda A + (1 - \lambda)A_S \text{ where } \lambda \sim \text{Uniform}(0, \alpha). \quad (17)$$

This post-synthesis mixup can address our desideratum **C3** that we discussed before, that is, the generated samples for the target domain need to be distinguishable by class to solve the DG problem. Post-synthesis mixup, ensures synthesised amplitude images lie closer to the amplitude manifold of the source data, ensuring the model can solve the classification problem.

**Dense model averaging (SWAD).** We found that the DG training becomes more stable and the target-domain test performance becomes more consistent when we use the dense model averaging strategy SWAD (Cha et al., 2021). We adopt the SWAD model averaging for the variational and model parameters $(\lambda, \theta)$ while the generator network is not averaged.

**Amplitude image structures.** From the definition of the Fourier transform, the frequency domain function should be even-conjugate, i.e., $F(-u, -v) = \overline{F(u, v)}$, for the real-valued images. This implies that amplitude images are symmetric. Conversely, if the amplitude images are symmetric, inverse Fourier transform returns real-valued signals. Thus when generating amplitude images, we only generate the non-redundant part (frequencies) of the amplitude images. Also, the amplitude should be non-negative. We keep these constraints in mind when designing the generator network.

## 4 RELATED WORK

**MCD.** Several studies have used the MCD principle for domain adaptation, to align a source model to unlabeled target data (Saito et al., 2018; Kim et al., 2019; Lu et al., 2020). We uniquely exploit the MCD principle for the DG problem, in the absence of target data, by using MCD to synthesise worst-case target domain data, as well as to adapt the model to that synthesised domain.

**Augmentation approaches to DG.** Several DG approaches have been proposed based on data augmentation. Existing approaches either define augmentation heuristics (Zhou et al., 2021b; Xu et al., 2021), or exploit *domain* adversarial learning – i.e., confusing a *domain classifier* (Shankar et al., 2018; Zhou et al., 2020). Our adversarial learning is based on the much stronger (S)MCD principle that confuses a category classifier. This provides much harder examples for robust learning, while our Fourier amplitude synthesis ensures the examples are actually recognisable.

**Alignment approaches to DG.** Several approaches to DG are based on aligning between multiple source domains (Sun & Saenko, 2016; Ganin et al., 2016; Li et al., 2018c;b), under the assumption that a common feature across all source domains will be good for a held out target domain. Differently, we use the MCD principle to robustify our source trained model by aligning it with the synthesised worst-case target domain.

## 5 EXPERIMENTS

We test our approach on the DomainBed benchmark (Gulrajani & Lopez-Paz, 2021), including: **PACS** (Li et al., 2017), **VLCS** (Fang et al., 2013), **OfficeHome** (Venkateswara et al., 2017), **TerraIncognita** (Beery et al., 2018), and **DomainNet** (Peng et al., 2019). For each dataset, we adopt the standard leave-one-domain-out source/target domain splits. The overall training/test protocols are similar to (Gulrajani & Lopez-Paz, 2021; Cha et al., 2021). We use the ResNet-50 (He et al., 2016) as our feature extractor backbone, which is initialised by the pretrained weights on ImageNet (Deng et al., 2009). For the generator network, we found that a linear model performed the best for the noise dimension 100. Our model is trained by the Adam optimiser (Kingma & Ba, 2015) on machines with single Tesla V100 GPUs. The hyperparameters introduced in our model (e.g., SMCD trade-off $\eta$) and the general ones (e.g., learning rate, SWAD regime hyperparameters, maximum numbers of iterations) are chosen by grid search on the validation set according to the DomainBed protocol (Gulrajani & Lopez-Paz, 2021). For instance, $\eta = 0.1$ for all datasets. The implementation details including chosen hyperparameters can be found in Appendix A.1.

### 5.1 MAIN RESULTS

The test accuracies averaged over target domains are summarised in Table 1, where the results for individual target domains are reported in Appendix A.2. The proposed approach performs the best

Table 1: Average accuracies on DomainBed datasets. Note: $^\dagger$ indicates that the results are excerpted from the published papers or (Gulrajani & Lopez-Paz, 2021). Our own runs are reported without $^\dagger$. Note that FACT (Xu et al., 2021) adopted a slightly different data/domain split protocol from DomainBed's, explaining discrepancy on PACS.

| Algorithm | PACS | VLCS | OfficeHome | TerraInc. | DomainNet | Avg. |
|---|---|---|---|---|---|---|
| ERM (Cha et al., 2021)$^\dagger$ | 84.2 | 77.3 | 67.6 | 47.8 | 44.0 | 64.2 |
| IRM (Arjovsky et al., 2019)$^\dagger$ | 83.5 | 78.6 | 64.3 | 47.6 | 33.9 | 61.6 |
| GroupDRO (Sagawa et al., 2020)$^\dagger$ | 84.4 | 76.7 | 66.0 | 43.2 | 33.3 | 60.7 |
| I-Mixup (Xu et al., 2020; Yan et al., 2020; Wang et al., 2020b)$^\dagger$ | 84.6 | 77.4 | 68.1 | 47.9 | 39.2 | 63.4 |
| MLDG (Li et al., 2018a)$^\dagger$ | 84.9 | 77.2 | 66.8 | 47.8 | 41.2 | 63.6 |
| CORAL (Sun & Saenko, 2016)$^\dagger$ | 86.2 | 78.8 | 68.7 | 47.7 | 41.5 | 64.5 |
| MMD (Li et al., 2018b)$^\dagger$ | 84.7 | 77.5 | 66.4 | 42.2 | 23.4 | 58.8 |
| DANN (Ganin et al., 2016)$^\dagger$ | 83.7 | 78.6 | 65.9 | 46.7 | 38.3 | 62.6 |
| CDANN (Li et al., 2018c)$^\dagger$ | 82.6 | 77.5 | 65.7 | 45.8 | 38.3 | 62.0 |
| MTL (Blanchard et al., 2021)$^\dagger$ | 84.6 | 77.2 | 66.4 | 45.6 | 40.6 | 62.9 |
| SagNet (Nam et al., 2021)$^\dagger$ | 86.3 | 77.8 | 68.1 | 48.6 | 40.3 | 64.2 |
| ARM (Zhang et al., 2020)$^\dagger$ | 85.1 | 77.6 | 64.8 | 45.5 | 35.5 | 61.7 |
| VREx (Krueger et al., 2020)$^\dagger$ | 84.9 | 78.3 | 66.4 | 46.4 | 33.6 | 61.9 |
| RSC (Huang et al., 2020)$^\dagger$ | 85.2 | 77.1 | 65.5 | 46.6 | 38.9 | 62.7 |
| Mixstyle (Zhou et al., 2021b)$^\dagger$ | 85.2 | 77.9 | 60.4 | 44.0 | 34.0 | 60.3 |
| FACT (Xu et al., 2021)$^\dagger$ | 88.2 | — | 66.6 | — | — | — |
| FACT (Xu et al., 2021) | 86.4 | 76.6 | 66.6 | 45.4 | 42.6 | 63.5 |
| Amp-Mixup (Xu et al., 2021) | 84.7 | 75.9 | 64.0 | 46.8 | 42.0 | 62.7 |
| SWAD (Cha et al., 2021)$^\dagger$ | 88.1 | 79.1 | 70.6 | 50.0 | 46.5 | 66.9 |
| FACT+SWAD | 88.1 | 77.7 | 70.6 | 51.0 | 46.7 | 66.8 |
| Amp-Mixup+SWAD | 88.1 | 78.2 | 70.3 | 51.2 | 46.4 | 66.8 |
| (Proposed) AGFA | **89.3** | **79.5** | **71.5** | **52.4** | **47.1** | **68.0** |

for all datasets among the competitors, and the difference from the second best model (SWAD) is significant (about $1.1\%$ margin). We particularly contrast with two recent approaches: **SWAD** (Cha et al., 2021) that adopts the dense model averaging with the simple ERM loss and **FACT** (Xu et al., 2021) that uses the Fourier amplitude mixup as means of data augmentation with additional student-teacher regularisation.

First, SWAD (Cha et al., 2021) is the second best model in Table 1, implying that the simple ERM loss combined with the dense model averaging that seeks for flat minima is quite effective, also observed previously (Gulrajani & Lopez-Paz, 2021). FACT (Xu et al., 2021) utilises the Fourier amplitude spectra similar to our approach, but their main focus is *data augmentation*, producing more training images by amplitude mixup of source domain images. FACT also adopted the so-called *teacher co-regularisation* which forces the orders of the class prediction logits to be consistent between teacher and student models on the amplitude-mixup data. To disentangle the impact of these two components in FACT, we ran a model called **Amp-Mixup** that is simply FACT without teacher co-regularisation. The teacher co-regularisation yields further improvement in the average accuracy (FACT > Amp-Mixup in the last column of Table 1), verifying the claim in (Xu et al., 2021), although FACT is slightly worse than Amp-Mixup on VLCS and TerraIncognita.

We also modified FACT and Amp-Mixup models by incorporating the SWAD model averaging (FACT+SWAD and Amp-Mixup+SWAD in the table). Clearly they perform even better in combination with SWAD. Since Amp-Mixup+SWAD can be seen as dropping the teacher regularisation and adopting the SWAD (regularisation) strategy instead, we can say that SWAD is more effective regularisation than student-teacher. Nevertheless, despite the utilisation of amplitude-mixup augmentation, it appears that FACT and Amp-Mixup have little improvement over the ERM loss even when the SWAD strategy is used. This signifies the effect of the adversarial Fourier-based target domain generation in our approach which exhibits significant improvement over ERM and SWAD.

## 5.2 FURTHER ANALYSIS

**Sensitivity to $\eta$ (SMCD strength).** We analyze sensitivity of the target domain generalisation performance to the SMCD trade-off hyperparameter $\eta$. We run our algorithm with different values of $\eta$. The results are shown in Fig. 3. Note that $\eta = 0$ ignores the SMCD loss term (thus generator has no influence on the model training), which corresponds to the ERM approach. The test accuracy of the proposed approach remains significantly better than ERM/SWAD for all those $\eta$ with moderate variations around the best value. See Appendix A.2 for the results on individual target domains.

**Sensitivity to $\alpha$ (post-synthesis mixup strength).** We mix up the generated amplitude images and the source domain images as in (17) to make the adversarial target domain classification task solvable. The task becomes easier for small $\alpha$ (less impact of the generated amplitudes), and vice versa. Note

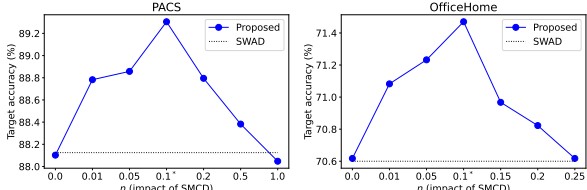

Figure 3: Sensitivity to $\eta$ (SMCD trade-off) on PACS and OfficeHome.

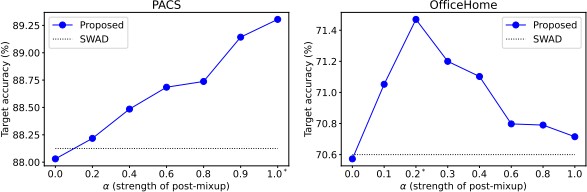

Figure 4: Sensitivity to $\alpha$ (post-mixup strength) on PACS and OfficeHome.

Table 2: Ablation study: 1) unsupervised MCD (instead of SMCD), 2) without post-mixup, 3) without SWAD, and 4) pixel-based target image generation (instead of amplitude generation).

|  | Art | Cartoon | Photo | Sketch | Average |
|---|---|---|---|---|---|
| Unsupervised MCD | $88.94 \pm 0.23$ | $83.83 \pm 0.19$ | $97.27 \pm 0.10$ | $81.77 \pm 0.36$ | 87.95 |
| Without post-mixup | $88.90 \pm 0.16$ | $81.80 \pm 0.17$ | $97.43 \pm 0.14$ | $80.86 \pm 0.31$ | 87.25 |
| Without SWAD | $84.20 \pm 0.68$ | $81.56 \pm 0.55$ | $94.83 \pm 0.12$ | $79.28 \pm 0.94$ | 84.97 |
| Pixel-based generation | $88.85 \pm 0.15$ | $83.62 \pm 0.26$ | $97.23 \pm 0.15$ | $82.10 \pm 0.63$ | 87.95 |
| (Proposed) AGFA | $\mathbf{89.80 \pm 0.34}$ | $\mathbf{85.16 \pm 0.65}$ | $\mathbf{97.59 \pm 0.27}$ | $\mathbf{84.67 \pm 0.82}$ | $\mathbf{89.30}$ |

that $\alpha = 0$ ignores generated amplitude images completely in post-mixup, and the training becomes close to ERM learning where the only difference is that we utilise more basic augmentation (e.g., flip, rotation, color jittering). As shown in Fig. 4, the target test performance is not very sensitive around the best selected hyperparameters. See also ablation study results on the impact of post-mixup below.

**Impact of SMCD (vs. unsupervised MCD).** We verify the positive effect of the proposed *supervised MCD* loss (SMCD in (16)) that exploits the induced target domain class labels, compared to the conventional (unsupervised) MCD loss (14) without using the target class labels. The result in Table 2 supports our claim that exploiting target class labels induced from the phase information is quite effective, improving the target generalisation performance.

**Impact of post-synthesis mixup.** We argued that our post-synthesis mixup of the generated amplitude images makes the class prediction task easier for the generated target domain, for the solvability of the DG problem. To verify this, we compare two models, with and without the post-mixup strategy in Table 2. The model trained with post-mixup performs better.

**Impact of SWAD.** We adopted the SWAD model averaging scheme (Cha et al., 2021) for improving generalisation performance. We verify the impact of the SWAD as in Table 2 where the model without SWAD has lower target test accuracy signifying the importance of the SWAD model averaging.

**Impact of amplitude generation.** The amplitude image generation in our adversarial MCD learning allows us to separate the phase and amplitude images and exploit the class labels induced by the phase information. However, one may be curious about how the model would work if we instead generate full images without phase/amplitude separation in an adversarial way. That is, we adopt a pixel-based adversarial image generator, and in turn replace our SMCD by the conventional MCD loss (since there are no class labels inducible in this strategy). We consider two generator architectures: linear (from 100-dim input noise to full image pixels) and nonlinear (a fully connected network with one hidden layer of 100 units), where the former slightly performs better. Table 2 shows that this pixel-based target image generation underperforms our amplitude generation.

## 6 CONCLUSION

We tried to address the domain generalisation problem from the perspective of maximum classifier discrepancy: Improving robustness by synthesising a worst-case target domain for learning, and training the model to be robust to that domain with the (S)MCD objective. To provide an approximation to style and content separation for synthesis, the worst-case domain is synthesised in Fourier amplitude space. Our results provide a clear improvement on the state-of-the-arts on the challenging DomainBed benchmark suite.

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

## A  APPENDIX

The Appendix consists of the following contents:

- Implementation Details (Sec. A.1)
- Full Results (Sec. A.2)
- Derivation of ELBO in Variational Inference (Sec. A.3)
- Additional Experimental Results (Sec. A.4)

### A.1  IMPLEMENTATION DETAILS

We adopt the ResNet50 (He et al., 2016) architecture (removing the final classification layer) as the feature extractor network. For the amplitude generator network, we have tested several fully-connected network architectures with different numbers of hidden layers and hidden units, and the simple linear network peformed the best. The input noise dimension for the generator is chosen as 100. The covariance matrices of the variational parameters are restricted to be diagonal. The number of MC samples from $Q_\lambda(W)$ in the ELBO optimisation is chosen as 50.

The optimisation hyperparameters are chosen by the same strategy as (Cha et al., 2021), where we employ the Adam optimiser (Kingma & Ba, 2015) with learning rate $5 \times 10^{-5}$, and no dropout, weight decay used. The batch size was 32 (for each training domain) in ERM/SWAD (Cha et al., 2021), but we halved it in our model since the remaining half are constructed by the adversarial target generation. The standard basic data augmentation is also applied to the input images. Following the suggestion from (Cha et al., 2021), we run our model up to 5000 iterations for all datasets except for DomainNet. But the algorithm may stop earlier before the maximum iterations if SWAD termination condition is met (See Sec. A.1.1 below). Since DomainNet is a large-scale dataset, and it is required to have a even larger number of iterations to go through the entire data at least several times. In (Cha et al., 2021), they used 15000 iterations which roughly corresponds to 3 to 10 data epochs. In our model, since we halved the number of input images in the batch, in order to have the same training epochs as (Cha et al., 2021), we increase it up to 30000 iterations for DomainNet. The details of the SWAD implementation follows in the next section.

#### A.1.1  SWAD MODEL AVERAGING

We adopt the SWAD model averaging strategy (Cha et al., 2021) to have a more robust model that is less affected by overfitting. We apply the SWAD to the feature extractor network parameters $\theta$ and the variational parameters $\lambda$, but not the adversarial generator network. Since SWAD is an important component in our model, we provide more details here.

SWAD is motivated from *stochastic weight averaging* (SWA) (Izmailov et al., 2018), however, unlike SWA's model averaging for every *epoch*, SWAD takes dense model averaging for every (batch) *iteration*. A key component of the SWAD algorithm is to determine the model averaging regime, the interval of iterations for which the model averaging is performed. This regime is aimed to avoid overfitting, and known as overfit-aware model averaging. The regime is specified by the start and end iteration numbers, $t_s$ and $t_e$, respectively, and we take model averaging for iterations $t \in [t_s, t_e]$, that is,

$$\theta_{SWAD} = \frac{1}{t_e - t_s + 1} \sum_{t=t_s}^{t_e} \theta^t, \quad \lambda_{SWAD} = \frac{1}{t_e - t_s + 1} \sum_{t=t_s}^{t_e} \lambda^t, \qquad (18)$$

where $\theta^t$ and $\lambda^t$ are the model parameters after iteration $t$. Here $(\theta_{SWAD}, \lambda_{SWAD})$ are the final model parameters returned by the training algorithm.

Now we describe how the regime is determined. Ideally, we expect the intermediate models during the interval $[t_s, t_e]$ to be overfit-free, having high generalisation performance. To this end, we evaluate the model on the validation set (held out from the source domain training data), and denote the validation loss of the $t$-th model by $l_{val}^t$. Then the start iteration of the regime, $t_s$ is determined by the first $t$ where the validation loss is not improved for the next $N_s$ iterations (e.g., $N_s = 3$). That is,

$$t_s = \min\{t - N_s + 1 \mid l_{val}^{t-N_s+1} \le l_{val}^t, l_{val}^{t-1}, \ldots, l_{val}^{t-N_s+1}\}. \qquad (19)$$

---

**Algorithm 1** AGFA Algorithm with SWAD Model Averaging.

---

**Input:** Source data $S$, SMCD trade-off $\eta$, post-mixup $\alpha$, and learning rate $\gamma$, and
      SWAD hyperparameters $N_s$, $N_e$, $r$.
**Initialise:** $\theta$ (feature extractor), $\lambda$ (variational parameters), and $\nu$ (generator).
      (flag) SWAD-Regime-Entered $\leftarrow FALSE$, (iteration) $t \leftarrow 0$.
**Repeat:**
   0. Sample a minibatch $S_B = \{(x_i^S, y_i^S)\}_{i=1}^n$ from $S$.
   1. Prepare $\{(A_i^S, P_i^S)\}_{i=1}^n$ by Fourier transform $A_i^S \angle P_i^S = \mathcal{F}(x_i^S)$.
   2. Generate amplitude images $A_i^G = G_\nu(\epsilon_i)$, $\epsilon_i \sim \mathcal{N}(0, I)$ for $i = 1, \ldots, n$.
   3. Post-mixup: $A_i^G \leftarrow \lambda A_i^G + (1 - \lambda)A_i^S$, $\lambda \sim \text{Uniform}(0, \alpha)$.
   4. Construct a target batch $T_B = \{(x_i^T, y_i^T)\}_{i=1}^n$: $x_i^T = \mathcal{F}^{-1}(A_i^G \angle P_i^S)$, $y_i^T = y_i^S$.
   5. Evaluate $\mathcal{L}_{model} := -\text{ELBO}(\lambda, \theta; S_B) + \eta \text{SMCD}(\theta; T_B)$.
   6. Update the model and variational parameters: $(\lambda, \theta) \leftarrow (\lambda, \theta) - \gamma \nabla_{(\lambda, \theta)} \mathcal{L}_{model}$.
   7. Evaluate $\mathcal{L}_{gen} := -\text{SMCD}(\theta; T_B)$.
   8. Update the generator network: $\nu \leftarrow \nu - \gamma \nabla_\nu \mathcal{L}_{gen}$.
   9. (SWAD procedure)
      $t \leftarrow t + 1$, $(\lambda^t, \theta^t) \leftarrow (\lambda, \theta)$.
      If SWAD-Regime-Entered $== FALSE$:
         If $l_{val}^{t-N_s+1} = \min_{0 \le t' < N_s} l_{val}^{t-t'}$:
            $t_s \leftarrow t - N_s + 1$, $\bar{l}_{val} \leftarrow \frac{1}{N_s} \sum_{t'=0}^{N_s-1} l_{val}^{t-t'}$.
            SWAD-Regime-Entered $\leftarrow True$.
      Else:
         If $r \cdot \bar{l}_{val} < \min_{0 \le t' < N_e} l_{val}^{t-t'}$:
            $t_e \leftarrow t - N_e$.
            Return $\theta_{SWAD} = \frac{1}{t_e - t_s + 1} \sum_{t=t_s}^{t_e} \theta^t$, $\lambda_{SWAD} = \frac{1}{t_e - t_s + 1} \sum_{t=t_s}^{t_e} \lambda^t$.

---

Once we find $t_s$, we compute the (average) starting validation loss,

$$\bar{l}_{val} = \frac{\sum_{t=t_s}^{t_s+N_s-1} l_{val}^t}{N_s}, \tag{20}$$

which is used as a reference when we decide the end iteration $t_e$. As we enter the regime, we start model averaging every iteration. To determine when to stop, we inspect the validation losses to see if the model starts overfitting. Specifically, if the validation losses are consecutively greater than $\bar{l}_{val}$ by some margin, we regard it as overfit signal. That is,

$$t_e = \min\{t - N_e \mid l_{val}^t, l_{val}^{t-1}, \ldots, l_{val}^{t-N_e+1} > r \cdot \bar{l}_{val}\}, \tag{21}$$

where $r$ and $N_e$ are user-driven hyperparameters (e.g., $r = 1.3$, $N_e = 6$).

The pseudo code of our AGFA algorithm with the SWAD strategy is summarised in Alg. 1. There are three hyperparameters in SWAD, $(N_s, N_e, r)$, and following (Cha et al., 2021), we use $N_s = 3$, $N_e = 6$, $r = 1.3$ for all datasets in DomainBed except $r = 1.2$ for VLCS. One technical issue is that evaluating the validation loss every iteration is computationally demanding. Similarly as (Cha et al., 2021), we compute the validation loss at every $V$-th iterations (e.g., $V = 50$ for VLCS, $V = 500$ for DomainNet, and $V = 100$ for the rest) although the model averaging is still performed every iteration. Accordingly, the equations (19), (20), and (21) need to be changed where essentially all iteration numbers in those equations should be changed to multiples of $V$. The model averaging in Alg. 1 is implemented by the running (online) average and the use of (FIFO) queue data structures similarly as (Cha et al., 2021), which does not incur significant extra computational overhead.

## A.2 FULL RESULTS

The full results (test errors on individual target domains) on DomainBed datasets are summarised in Table 3 (PACS), Table 4 (VLCS), Table 5 (OfficeHome), Table 6 (TerraIncognita), and Table 7 (DomainNet).

We also show the full results of the sensitivity analysis in Table 8 (the SMCD loss trade-off $\eta$) and Table 9 (the post-synthesis mixup strength $\alpha$). Moreover, we visualise in Fig. 5 the ablation study

Table 3: Average accuracies on PACS. Note: † indicates that the results are excerpted from the published papers or (Gulrajani & Lopez-Paz, 2021). Our own runs are reported without †. FACT (Xu et al., 2021) adopted a slightly different data/domain split from DomainBed's, explaining discrepancy.

| Algorithm | A | C | P | S | Avg |
|---|---|---|---|---|---|
| MASF (Dou et al., 2019)† | 82.9 | 80.5 | 95.0 | 72.3 | 82.7 |
| DMG (Chattopadhyay et al., 2020)† | 82.6 | 78.1 | 94.5 | 78.3 | 83.4 |
| MetaReg (Balaji et al., 2018)† | 87.2 | 79.2 | 97.6 | 70.3 | 83.6 |
| ER (Zhao et al., 2020)† | 87.5 | 79.3 | **98.3** | 76.3 | 85.3 |
| pAdaIN (Nuriel et al., 2021)† | 85.8 | 81.1 | 97.2 | 77.4 | 85.4 |
| EISNet (Wang et al., 2020a)† | 86.6 | 81.5 | 97.1 | 78.1 | 85.8 |
| DSON (Seo et al., 2020)† | 87.0 | 80.6 | 96.0 | 82.9 | 86.6 |
| ERM (Cha et al., 2021)† | $85.7 \pm 0.6$ | $77.1 \pm 0.8$ | $97.4 \pm 0.4$ | $76.6 \pm 0.7$ | 84.2 |
| IRM (Arjovsky et al., 2019)† | $84.8 \pm 1.3$ | $76.4 \pm 1.1$ | $96.7 \pm 0.6$ | $76.1 \pm 1.0$ | 83.5 |
| GroupDRO (Sagawa et al., 2020)† | $83.5 \pm 0.9$ | $79.1 \pm 0.6$ | $96.7 \pm 0.3$ | $78.3 \pm 2.0$ | 84.4 |
| I-Mixup (Xu et al., 2020; Yan et al., 2020; Wang et al., 2020b)† | $86.1 \pm 0.5$ | $78.9 \pm 0.8$ | $97.6 \pm 0.1$ | $75.8 \pm 1.8$ | 84.6 |
| MLDG (Li et al., 2018a)† | $85.5 \pm 1.4$ | $80.1 \pm 1.7$ | $97.4 \pm 0.3$ | $76.6 \pm 1.1$ | 84.9 |
| CORAL (Sun & Saenko, 2016)† | $88.3 \pm 0.2$ | $80.0 \pm 0.5$ | $97.5 \pm 0.3$ | $78.8 \pm 1.3$ | 86.2 |
| MMD (Li et al., 2018b)† | $86.1 \pm 1.4$ | $79.4 \pm 0.9$ | $96.6 \pm 0.2$ | $76.5 \pm 0.5$ | 84.7 |
| DANN (Ganin et al., 2016)† | $86.4 \pm 0.8$ | $77.4 \pm 0.8$ | $97.3 \pm 0.4$ | $73.5 \pm 2.3$ | 83.7 |
| CDANN (Li et al., 2018c)† | $84.6 \pm 1.8$ | $75.5 \pm 0.9$ | $96.8 \pm 0.3$ | $73.5 \pm 0.6$ | 82.6 |
| MTL (Blanchard et al., 2021)† | $87.5 \pm 0.8$ | $77.1 \pm 0.5$ | $96.4 \pm 0.8$ | $77.3 \pm 1.8$ | 84.6 |
| SagNet (Nam et al., 2021)† | $87.4 \pm 1.0$ | $80.7 \pm 0.6$ | $97.1 \pm 0.1$ | $80.0 \pm 0.4$ | 86.3 |
| ARM (Zhang et al., 2020)† | $86.8 \pm 0.6$ | $76.8 \pm 0.5$ | $97.4 \pm 0.3$ | $79.3 \pm 1.2$ | 85.1 |
| VREx (Krueger et al., 2020)† | $86.0 \pm 1.6$ | $79.1 \pm 0.6$ | $96.9 \pm 0.5$ | $77.7 \pm 1.7$ | 84.9 |
| RSC (Huang et al., 2020)† | $85.4 \pm 0.8$ | $79.7 \pm 1.8$ | $97.6 \pm 0.3$ | $78.2 \pm 1.2$ | 85.2 |
| Mixstyle (Zhou et al., 2021b)† | $86.8 \pm 0.5$ | $79.0 \pm 1.4$ | $96.6 \pm 0.1$ | $78.5 \pm 2.3$ | 85.2 |
| FACT (Xu et al., 2021)† | $89.6 \pm 0.5$ | $81.8 \pm 0.2$ | $96.8 \pm 0.1$ | $84.5 \pm 0.8$ | 88.2 |
| FACT (Xu et al., 2021) | $87.8 \pm 0.2$ | $80.5 \pm 1.1$ | $96.2 \pm 0.2$ | $81.2 \pm 0.6$ | 86.4 |
| Amp-Mixup (Xu et al., 2021) | $84.7 \pm 0.6$ | $81.0 \pm 1.1$ | $95.0 \pm 0.2$ | $78.1 \pm 1.0$ | 84.7 |
| SWAD (Cha et al., 2021)† | $89.3 \pm 0.2$ | $83.4 \pm 0.6$ | $97.3 \pm 0.3$ | $82.5 \pm 0.5$ | 88.1 |
| FACT+SWAD | $89.6 \pm 0.8$ | $82.5 \pm 0.3$ | $96.6 \pm 0.2$ | $83.8 \pm 0.8$ | 88.1 |
| Amp-Mixup+SWAD | $88.7 \pm 0.1$ | $83.2 \pm 0.4$ | $96.4 \pm 0.1$ | $84.1 \pm 0.5$ | 88.1 |
| (Proposed) AGFA | $\mathbf{89.8 \pm 0.3}$ | $\mathbf{85.2 \pm 0.6}$ | $97.6 \pm 0.3$ | $\mathbf{84.7 \pm 0.8}$ | **89.3** |

results for the four different modeling choices: 1) Impact of SMCD (vs. conventional unsupervised MCD), 2) Impact of post-synthesis mixup, 3) Impact of SWAD, and 4) Impact of amplitude generation (vs. pixel-based image generation). For the pixel-based image generation, we consider two generator architectures: linear (from 100-dim input noise to full image pixels) and nonlinear (a fully connected network with one hidden layer of 100 units).

**Visualisation of generated adversarial images.** We visualise in Fig. 6 some synthesised amplitude images and constructed target domain images from the learned model on the PACS dataset. Although the generated amplitude images visually look like random noise, they appear to have the effect of attenuating high frequency spectra (shown as darker pixels in the fifth column) when combined with the source domain amplitude images by post-mixup. The constructed images from the generated amplitude images alone without post-mixup (sixth column) look a lot like edge detection maps, whereas the post-mixup constructed ones (seventh column) remain visually similar to the original source domain images, promoting DG solvability.

## A.3 DERIVATION OF ELBO IN VARIATIONAL INFERENCE

We derive the evidence lower bound (ELBO) in (11) in the main paper. To enforce $Q_\lambda(W) \approx P(W|S, \theta)$, we minimise their KL divergence,

$$\mathrm{KL}\big(Q_\lambda(W)||P(W|S,\theta)\big) = \mathbb{E}_{Q_\lambda(W)}\left[\log \frac{Q_\lambda(W)}{P(W|S,\theta)}\right] \tag{22}$$

$$= \mathbb{E}_{Q_\lambda(W)}\left[\log \frac{Q_\lambda(W)P(S|\theta)}{P(S|W,\theta)P(W)}\right] \tag{23}$$

$$= \log P(S|\theta) - \mathbb{E}_{Q_\lambda(W)}\big[\log P(S|W,\theta)\big] + \mathbb{E}_{Q_\lambda(W)}\left[\log \frac{Q_\lambda(W)}{P(W)}\right] \tag{24}$$

$$= \log P(S|\theta) - \mathbb{E}_{Q_\lambda(W)}\big[\log P(S|W,\theta)\big] + \mathrm{KL}\big(Q_\lambda(W)||P(W)\big). \tag{25}$$

$$= \log P(S|\theta) - \sum_{(x,y)\sim S} \mathbb{E}_{Q_\lambda(W)}\big[\log P(y|x,W,\theta)\big] + \mathrm{KL}\big(Q_\lambda(W)||P(W)\big). \tag{26}$$

Table 4: Average accuracies on VLCS. The same interpretation as Table 3.

| Algorithm | C | L | S | V | Avg |
|---|---|---|---|---|---|
| ERM (Cha et al., 2021)[†] | $98.0 \pm 0.3$ | $64.7 \pm 1.2$ | $71.4 \pm 1.2$ | $75.2 \pm 1.6$ | 77.3 |
| IRM (Arjovsky et al., 2019)[†] | $98.6 \pm 0.1$ | $64.9 \pm 0.9$ | $73.4 \pm 0.6$ | $77.3 \pm 0.9$ | 78.6 |
| GroupDRO (Sagawa et al., 2020)[†] | $97.3 \pm 0.3$ | $63.4 \pm 0.9$ | $69.5 \pm 0.8$ | $76.7 \pm 0.7$ | 76.7 |
| I-Mixup (Xu et al., 2020; Yan et al., 2020; Wang et al., 2020b)[†] | $98.3 \pm 0.6$ | $64.8 \pm 1.0$ | $72.1 \pm 0.5$ | $74.3 \pm 0.8$ | 77.4 |
| MLDG (Li et al., 2018a)[†] | $97.4 \pm 0.2$ | $65.2 \pm 0.7$ | $71.0 \pm 1.4$ | $75.3 \pm 1.0$ | 77.2 |
| CORAL (Sun & Saenko, 2016)[†] | $98.3 \pm 0.1$ | $\mathbf{66.1 \pm 1.2}$ | $73.4 \pm 0.3$ | $77.5 \pm 1.2$ | 78.8 |
| MMD (Li et al., 2018b)[†] | $97.7 \pm 0.1$ | $64.0 \pm 1.1$ | $72.8 \pm 0.2$ | $75.3 \pm 3.3$ | 77.5 |
| DANN (Ganin et al., 2016)[†] | $\mathbf{99.0 \pm 0.3}$ | $65.1 \pm 1.4$ | $73.1 \pm 0.3$ | $77.2 \pm 0.6$ | 78.6 |
| CDANN (Li et al., 2018c)[†] | $97.1 \pm 0.3$ | $65.1 \pm 1.2$ | $70.7 \pm 0.8$ | $77.1 \pm 1.5$ | 77.5 |
| MTL (Blanchard et al., 2021)[†] | $97.8 \pm 0.4$ | $64.3 \pm 0.3$ | $71.5 \pm 0.7$ | $75.3 \pm 1.7$ | 77.2 |
| SagNet (Nam et al., 2021)[†] | $97.9 \pm 0.4$ | $64.5 \pm 0.5$ | $71.4 \pm 1.3$ | $77.5 \pm 0.5$ | 77.8 |
| ARM (Zhang et al., 2020)[†] | $98.7 \pm 0.2$ | $63.6 \pm 0.7$ | $71.3 \pm 1.2$ | $76.7 \pm 0.6$ | 77.6 |
| VREx (Krueger et al., 2020)[†] | $98.4 \pm 0.3$ | $64.4 \pm 1.4$ | $74.1 \pm 0.4$ | $76.2 \pm 1.3$ | 78.3 |
| RSC (Huang et al., 2020)[†] | $97.9 \pm 0.1$ | $62.5 \pm 0.7$ | $72.3 \pm 1.2$ | $75.6 \pm 0.8$ | 77.1 |
| Mixstyle (Zhou et al., 2021b)[†] | $98.6 \pm 0.3$ | $64.5 \pm 1.1$ | $72.6 \pm 0.5$ | $75.7 \pm 1.7$ | 77.9 |
| FACT (Xu et al., 2021) | $97.6 \pm 0.1$ | $65.5 \pm 0.5$ | $69.2 \pm 0.8$ | $73.9 \pm 0.7$ | 76.6 |
| Amp-Mixup (Xu et al., 2021) | $97.4 \pm 0.7$ | $65.6 \pm 0.3$ | $70.5 \pm 0.9$ | $70.1 \pm 0.8$ | 75.9 |
| SWAD (Cha et al., 2021)[†] | $98.8 \pm 0.1$ | $63.3 \pm 0.3$ | $75.3 \pm 0.5$ | $\mathbf{79.2 \pm 0.6}$ | 79.1 |
| FACT+SWAD | $98.4 \pm 0.1$ | $63.1 \pm 0.3$ | $72.4 \pm 0.5$ | $77.0 \pm 0.4$ | 77.7 |
| Amp-Mixup+SWAD | $98.7 \pm 0.1$ | $63.9 \pm 0.5$ | $73.5 \pm 0.2$ | $76.7 \pm 0.2$ | 78.2 |
| (Proposed) AGFA | $\mathbf{99.0 \pm 0.1}$ | $64.5 \pm 0.6$ | $\mathbf{75.4 \pm 0.3}$ | $78.9 \pm 0.6$ | $\mathbf{79.5}$ |

Table 5: Average accuracies on OfficeHome. The same interpretation as Table 3.

| Algorithm | C | L | S | V | Avg |
|---|---|---|---|---|---|
| ERM (Cha et al., 2021)[†] | $63.1 \pm 0.3$ | $51.9 \pm 0.4$ | $77.2 \pm 0.5$ | $78.1 \pm 0.2$ | 67.6 |
| IRM (Arjovsky et al., 2019)[†] | $58.9 \pm 2.3$ | $52.2 \pm 1.6$ | $72.1 \pm 2.9$ | $74.0 \pm 2.5$ | 64.3 |
| GroupDRO (Sagawa et al., 2020)[†] | $60.4 \pm 0.7$ | $52.7 \pm 1.0$ | $75.0 \pm 0.7$ | $76.0 \pm 0.7$ | 66.0 |
| I-Mixup (Xu et al., 2020; Yan et al., 2020; Wang et al., 2020b)[†] | $62.4 \pm 0.8$ | $54.8 \pm 0.6$ | $76.9 \pm 0.3$ | $78.3 \pm 0.2$ | 68.1 |
| MLDG (Li et al., 2018a)[†] | $61.5 \pm 0.9$ | $53.2 \pm 0.6$ | $75.0 \pm 1.2$ | $77.5 \pm 0.4$ | 66.8 |
| CORAL (Sun & Saenko, 2016)[†] | $65.3 \pm 0.4$ | $54.4 \pm 0.5$ | $76.5 \pm 0.1$ | $78.4 \pm 0.5$ | 68.7 |
| MMD (Li et al., 2018b)[†] | $60.4 \pm 0.2$ | $53.3 \pm 0.3$ | $74.3 \pm 0.1$ | $77.4 \pm 0.6$ | 66.4 |
| DANN (Ganin et al., 2016)[†] | $59.9 \pm 1.3$ | $53.0 \pm 0.3$ | $73.6 \pm 0.7$ | $76.9 \pm 0.5$ | 65.9 |
| CDANN (Li et al., 2018c)[†] | $61.5 \pm 1.4$ | $50.4 \pm 2.4$ | $74.4 \pm 0.9$ | $76.6 \pm 0.8$ | 65.7 |
| MTL (Blanchard et al., 2021)[†] | $61.5 \pm 0.7$ | $52.4 \pm 0.6$ | $74.9 \pm 0.4$ | $76.8 \pm 0.4$ | 66.4 |
| SagNet (Nam et al., 2021)[†] | $63.4 \pm 0.2$ | $54.8 \pm 0.4$ | $75.8 \pm 0.4$ | $78.3 \pm 0.3$ | 68.1 |
| ARM (Zhang et al., 2020)[†] | $58.9 \pm 0.8$ | $51.0 \pm 0.5$ | $74.1 \pm 0.1$ | $75.2 \pm 0.3$ | 64.8 |
| VREx (Krueger et al., 2020)[†] | $60.7 \pm 0.9$ | $53.0 \pm 0.9$ | $75.3 \pm 0.1$ | $76.6 \pm 0.5$ | 66.4 |
| RSC (Huang et al., 2020)[†] | $60.7 \pm 1.4$ | $51.4 \pm 0.3$ | $74.8 \pm 1.1$ | $75.1 \pm 1.3$ | 65.5 |
| Mixstyle (Zhou et al., 2021b)[†] | $51.1 \pm 0.3$ | $53.2 \pm 0.4$ | $68.2 \pm 0.7$ | $69.2 \pm 0.6$ | 60.4 |
| FACT (Xu et al., 2021)[†] | $60.3 \pm 0.1$ | $54.9 \pm 0.4$ | $74.5 \pm 0.1$ | $76.6 \pm 0.1$ | 66.6 |
| FACT (Xu et al., 2021) | $61.2 \pm 0.1$ | $55.2 \pm 0.1$ | $74.0 \pm 0.2$ | $76.2 \pm 0.4$ | 66.6 |
| Amp-Mixup (Xu et al., 2021) | $57.1 \pm 0.3$ | $51.9 \pm 0.1$ | $72.5 \pm 0.3$ | $74.4 \pm 0.2$ | 64.0 |
| SWAD (Cha et al., 2021)[†] | $66.1 \pm 0.4$ | $57.7 \pm 0.4$ | $78.4 \pm 0.1$ | $80.2 \pm 0.2$ | 70.6 |
| FACT+SWAD | $66.4 \pm 0.2$ | $58.3 \pm 0.2$ | $78.0 \pm 0.1$ | $79.6 \pm 0.1$ | 70.6 |
| Amp-Mixup+SWAD | $65.9 \pm 0.2$ | $57.9 \pm 0.4$ | $77.8 \pm 0.2$ | $79.7 \pm 0.1$ | 70.3 |
| (Proposed) AGFA | $\mathbf{67.5 \pm 0.3}$ | $\mathbf{58.5 \pm 0.1}$ | $\mathbf{79.3 \pm 0.1}$ | $\mathbf{80.7 \pm 0.1}$ | $\mathbf{71.5}$ |

Since KL divergence is non-negative, re-arranging (26) yields:

$$\log P(S|\theta) \geq \sum_{(x,y)\sim S} \mathbb{E}_{Q_\lambda(W)} \big[ \log P(y|x, W, \theta) \big] - \mathrm{KL}\big(Q_\lambda(W)||P(W)\big), \qquad (27)$$

and the right hand side constitutes the ELBO.

## A.4    ADDITIONAL EXPERIMENTAL RESULTS

### A.4.1    RESULTS ON RESNET-18 BACKBONE

To test our approach on backbone networks other than ResNet-50, we run experiments with the ResNet-18 backbone on the PACS dataset. The results are summarised in Table 10. Compared to the recent approaches MixStyle (Zhou et al., 2021b) and EFDMix (Zhang et al., 2022), our approach AGFA again shows higher performance even with the smaller ResNet-18 backbone.

Table 6: Average accuracies on TerraIncognita. The same interpretation as Table 3.

| Algorithm | L100 | L38 | L43 | L46 | Avg |
|---|---|---|---|---|---|
| ERM (Cha et al., 2021)[†] | 54.3 ± 0.4 | 42.5 ± 0.7 | 55.6 ± 0.3 | 38.8 ± 2.5 | 47.8 |
| IRM (Arjovsky et al., 2019)[†] | 54.6 ± 1.3 | 39.8 ± 1.9 | 56.2 ± 1.8 | 39.6 ± 0.8 | 47.6 |
| GroupDRO (Sagawa et al., 2020)[†] | 41.2 ± 0.7 | 38.6 ± 2.1 | 56.7 ± 0.9 | 36.4 ± 2.1 | 43.2 |
| I-Mixup (Xu et al., 2020; Yan et al., 2020; Wang et al., 2020b)[†] | 59.6 ± 2.0 | 42.2 ± 1.4 | 55.9 ± 0.8 | 33.9 ± 1.4 | 47.9 |
| MLDG (Li et al., 2018a)[†] | 54.2 ± 3.0 | 44.3 ± 1.1 | 55.6 ± 0.3 | 36.9 ± 2.2 | 47.8 |
| CORAL (Sun & Saenko, 2016)[†] | 51.6 ± 2.4 | 42.2 ± 1.0 | 57.0 ± 1.0 | 39.8 ± 2.9 | 47.7 |
| MMD (Li et al., 2018b)[†] | 41.9 ± 3.0 | 34.8 ± 1.0 | 57.0 ± 1.9 | 35.2 ± 1.8 | 42.2 |
| DANN (Ganin et al., 2016)[†] | 51.1 ± 3.5 | 40.6 ± 0.6 | 57.4 ± 0.5 | 37.7 ± 1.8 | 46.7 |
| CDANN (Li et al., 2018c)[†] | 47.0 ± 1.9 | 41.3 ± 4.8 | 54.9 ± 1.7 | 39.8 ± 2.3 | 45.8 |
| MTL (Blanchard et al., 2021)[†] | 49.3 ± 1.2 | 39.6 ± 6.3 | 55.6 ± 1.1 | 37.8 ± 0.8 | 45.6 |
| SagNet (Nam et al., 2021)[†] | 53.0 ± 2.9 | 43.0 ± 2.5 | 57.9 ± 0.6 | 40.4 ± 1.3 | 48.6 |
| ARM (Zhang et al., 2020)[†] | 49.3 ± 0.7 | 38.3 ± 2.4 | 55.8 ± 0.8 | 38.7 ± 1.3 | 45.5 |
| VREx (Krueger et al., 2020)[†] | 48.2 ± 4.3 | 41.7 ± 1.3 | 56.8 ± 0.8 | 38.7 ± 3.1 | 46.4 |
| RSC (Huang et al., 2020)[†] | 50.2 ± 2.2 | 39.2 ± 1.4 | 56.3 ± 1.4 | 40.8 ± 0.6 | 46.6 |
| Mixstyle (Zhou et al., 2021b)[†] | 54.3 ± 1.1 | 34.1 ± 1.1 | 55.9 ± 1.1 | 31.7 ± 2.1 | 44.0 |
| FACT (Xu et al., 2021) | 52.4 ± 1.2 | 42.3 ± 1.0 | 55.5 ± 0.3 | 31.3 ± 0.9 | 45.4 |
| Amp-Mixup (Xu et al., 2021) | 56.0 ± 0.8 | 38.9 ± 0.7 | 56.9 ± 0.2 | 35.7 ± 0.8 | 46.8 |
| SWAD (Cha et al., 2021)[†] | 55.4 ± 0.0 | 44.9 ± 1.1 | 59.7 ± 0.4 | 39.9 ± 0.2 | 50.0 |
| FACT+SWAD | 57.0 ± 0.6 | **46.6 ± 1.1** | **60.3 ± 0.5** | 40.1 ± 0.3 | 51.0 |
| Amp-Mixup+SWAD | 56.6 ± 0.6 | 46.3 ± 0.3 | 60.2 ± 0.6 | 41.8 ± 0.4 | 51.2 |
| (Proposed) AGFA | **61.0 ± 0.3** | 46.2 ± 2.3 | **60.3 ± 0.7** | **42.3 ± 0.9** | **52.4** |

Table 7: Average accuracies on DomainNet. The same interpretation as Table 3.

| Algorithm | C | I | P | Q | R | S | Avg |
|---|---|---|---|---|---|---|---|
| DMG (Chattopadhyay et al., 2020)[†] | 65.2 | 22.2 | 50.0 | 15.7 | 59.6 | 49.0 | 43.6 |
| MetaReg (Balaji et al., 2018)[†] | 59.8 | **25.6** | 50.2 | 11.5 | 64.6 | 50.1 | 43.6 |
| ERM (Cha et al., 2021)[†] | 63.0 ± 0.2 | 21.2 ± 0.2 | 50.1 ± 0.4 | 13.9 ± 0.5 | 63.7 ± 0.2 | 52.0 ± 0.5 | 44.0 |
| IRM (Arjovsky et al., 2019)[†] | 48.5 ± 2.8 | 15.0 ± 1.5 | 38.3 ± 4.3 | 10.9 ± 0.5 | 48.2 ± 5.2 | 42.3 ± 3.1 | 33.9 |
| GroupDRO (Sagawa et al., 2020)[†] | 47.2 ± 0.5 | 17.5 ± 0.4 | 33.8 ± 0.5 | 9.3 ± 0.3 | 51.6 ± 0.4 | 40.1 ± 0.6 | 33.3 |
| I-Mixup (Citation as before) | 55.7 ± 0.3 | 18.5 ± 0.5 | 44.3 ± 0.5 | 12.5 ± 0.4 | 55.8 ± 0.3 | 48.2 ± 0.5 | 39.2 |
| MLDG (Li et al., 2018a)[†] | 59.1 ± 0.2 | 19.1 ± 0.3 | 45.8 ± 0.7 | 13.4 ± 0.3 | 59.6 ± 0.2 | 50.2 ± 0.4 | 41.2 |
| CORAL (Sun & Saenko, 2016)[†] | 59.2 ± 0.1 | 19.7 ± 0.2 | 46.6 ± 0.3 | 13.4 ± 0.4 | 59.8 ± 0.2 | 50.1 ± 0.6 | 41.5 |
| MMD (Li et al., 2018b)[†] | 32.1 ± 13.3 | 11.0 ± 4.6 | 26.8 ± 11.3 | 8.7 ± 2.1 | 32.7 ± 13.8 | 28.9 ± 11.9 | 23.4 |
| DANN (Ganin et al., 2016)[†] | 53.1 ± 0.2 | 18.3 ± 0.1 | 44.2 ± 0.7 | 11.8 ± 0.1 | 55.5 ± 0.4 | 46.8 ± 0.6 | 38.3 |
| CDANN (Li et al., 2018c)[†] | 54.6 ± 0.4 | 17.3 ± 0.1 | 43.7 ± 0.9 | 12.1 ± 0.7 | 56.2 ± 0.4 | 45.9 ± 0.5 | 38.3 |
| MTL (Blanchard et al., 2021)[†] | 57.9 ± 0.5 | 18.5 ± 0.4 | 46.0 ± 0.1 | 12.5 ± 0.1 | 59.5 ± 0.3 | 49.2 ± 0.1 | 40.6 |
| SagNet (Nam et al., 2021)[†] | 57.7 ± 0.3 | 19.0 ± 0.2 | 45.3 ± 0.3 | 12.7 ± 0.5 | 58.1 ± 0.5 | 48.8 ± 0.2 | 40.3 |
| ARM (Zhang et al., 2020)[†] | 49.7 ± 0.3 | 16.3 ± 0.5 | 40.9 ± 1.1 | 9.4 ± 0.1 | 53.4 ± 0.4 | 43.5 ± 0.4 | 35.5 |
| VREx (Krueger et al., 2020)[†] | 47.3 ± 3.5 | 16.0 ± 1.5 | 35.8 ± 4.6 | 10.9 ± 0.3 | 49.6 ± 4.9 | 42.0 ± 3.0 | 33.6 |
| RSC (Huang et al., 2020)[†] | 55.0 ± 1.2 | 18.3 ± 0.5 | 44.4 ± 0.6 | 12.2 ± 0.2 | 55.7 ± 0.7 | 47.8 ± 0.9 | 38.9 |
| Mixstyle (Zhou et al., 2021b)[†] | 51.9 ± 0.4 | 13.3 ± 0.2 | 37.0 ± 0.5 | 12.3 ± 0.1 | 46.1 ± 0.3 | 43.4 ± 0.4 | 34.0 |
| FACT (Xu et al., 2021) | 62.5 ± 0.3 | 19.4 ± 0.1 | 48.2 ± 0.4 | 13.9 ± 0.3 | 60.5 ± 0.7 | 51.0 ± 0.7 | 42.6 |
| Amp-Mixup (Xu et al., 2021) | 62.3 ± 0.1 | 19.0 ± 0.2 | 47.2 ± 0.4 | 12.9 ± 0.6 | 59.5 ± 0.3 | 51.0 ± 0.1 | 42.0 |
| SWAD (Cha et al., 2021)[†] | 66.0 ± 0.1 | 22.4 ± 0.3 | 53.5 ± 0.1 | 16.1 ± 0.2 | 65.8 ± 0.4 | 55.5 ± 0.3 | 46.5 |
| FACT+SWAD | 66.3 ± 0.1 | 22.7 ± 0.2 | 53.7 ± 0.1 | 16.3 ± 0.1 | 65.0 ± 0.6 | 55.9 ± 0.1 | 46.7 |
| Amp-Mixup+SWAD | 66.1 ± 0.1 | 22.4 ± 0.2 | 53.3 ± 0.1 | 16.2 ± 0.3 | 64.6 ± 0.5 | 55.6 ± 0.1 | 46.4 |
| (Proposed) AGFA | **66.7 ± 0.1** | 22.9 ± 0.2 | **54.0 ± 0.1** | **16.7 ± 0.2** | **65.9 ± 0.1** | **56.3 ± 0.1** | **47.1** |

### A.4.2 RESULTS ON COLOURED-MNIST AND ROTATED-MNIST

Although relatively smaller and easier datasets in the DomainBed benchmark, we also test our method on the Coloured-MNIST and Rotated-MNIST datasets. Following the experimental protocols including the four-layer ConvNet backbone as in (Gulrajani & Lopez-Paz, 2021), the test accuracies are reported in Table 11 (Colored-MNIST) and Table 12 (Rotated-MNIST). As shown, all approaches including ours perform equally well on these datasets.

### A.4.3 RESULTS ON SINGLE-SOURCE GENERALISATION

We have focused predominantly on the most popular leave-one-domain-out DG setting in our empirical study. Another reasonable experimental setting is single source generalisation setting: training on only one source domain and testing on the rest domains. Our single source domain results

Table 8: Sensitivity analysis on the SMCD loss trade off $\eta$ on PACS and OfficeHome.

(a) PACS

|  | Art | Cartoon | Photo | Sketch | Average |
|---|---|---|---|---|---|
| $\eta = 0.0$ | $89.08 \pm 0.14$ | $83.55 \pm 0.16$ | $97.23 \pm 0.19$ | $82.55 \pm 0.30$ | 88.10 |
| $\eta = 0.01$ | $89.24 \pm 0.26$ | $84.41 \pm 0.53$ | $97.17 \pm 0.07$ | $84.31 \pm 0.37$ | 88.78 |
| $\eta = 0.05$ | $89.52 \pm 0.26$ | $84.83 \pm 0.20$ | $97.33 \pm 0.17$ | $83.75 \pm 0.14$ | 88.86 |
| $\eta = 0.1$ | $\mathbf{89.80 \pm 0.34}$ | $\mathbf{85.16 \pm 0.65}$ | $\mathbf{97.59 \pm 0.27}$ | $\mathbf{84.67 \pm 0.82}$ | **89.30** |
| $\eta = 0.2$ | $89.40 \pm 0.60$ | $84.57 \pm 0.25$ | $97.33 \pm 0.15$ | $83.88 \pm 0.16$ | 88.80 |
| $\eta = 0.5$ | $89.00 \pm 0.14$ | $84.40 \pm 0.38$ | $97.03 \pm 0.17$ | $83.10 \pm 0.80$ | 88.38 |
| $\eta = 1.0$ | $89.11 \pm 0.36$ | $84.20 \pm 0.55$ | $96.49 \pm 0.22$ | $82.39 \pm 0.57$ | 88.05 |

(b) OfficeHome

|  | Art | Clipart | Product | Real | Average |
|---|---|---|---|---|---|
| $\eta = 0.0$ | $66.09 \pm 0.28$ | $57.72 \pm 0.34$ | $78.47 \pm 0.16$ | $80.19 \pm 0.11$ | 70.62 |
| $\eta = 0.01$ | $66.86 \pm 0.17$ | $58.43 \pm 0.34$ | $78.53 \pm 0.09$ | $80.51 \pm 0.31$ | 71.08 |
| $\eta = 0.05$ | $66.95 \pm 0.09$ | $\mathbf{58.56 \pm 0.32}$ | $78.96 \pm 0.28$ | $80.46 \pm 0.24$ | 71.23 |
| $\eta = 0.1$ | $\mathbf{67.46 \pm 0.28}$ | $58.45 \pm 0.13$ | $\mathbf{79.27 \pm 0.07}$ | $\mathbf{80.70 \pm 0.11}$ | **71.47** |
| $\eta = 0.15$ | $66.46 \pm 0.33$ | $58.31 \pm 0.22$ | $78.59 \pm 0.33$ | $80.51 \pm 0.09$ | 70.97 |
| $\eta = 0.2$ | $66.05 \pm 0.03$ | $58.10 \pm 0.11$ | $78.69 \pm 0.18$ | $80.45 \pm 0.06$ | 70.82 |
| $\eta = 0.25$ | $66.15 \pm 0.17$ | $58.29 \pm 0.36$ | $78.16 \pm 0.04$ | $79.87 \pm 0.21$ | 70.62 |

Table 9: Sensitivity analysis on the post-mixup trade off $\alpha$ on PACS and OfficeHome.

(a) PACS

|  | Art | Cartoon | Photo | Sketch | Average |
|---|---|---|---|---|---|
| $\alpha = 0.0$ | $89.29 \pm 0.37$ | $83.55 \pm 0.20$ | $97.11 \pm 0.07$ | $82.17 \pm 0.91$ | 88.03 |
| $\alpha = 0.2$ | $89.23 \pm 0.22$ | $83.80 \pm 0.21$ | $97.25 \pm 0.05$ | $82.59 \pm 0.84$ | 88.22 |
| $\alpha = 0.4$ | $89.37 \pm 0.17$ | $83.93 \pm 0.07$ | $97.27 \pm 0.10$ | $83.37 \pm 0.43$ | 88.49 |
| $\alpha = 0.6$ | $89.42 \pm 0.48$ | $84.30 \pm 0.16$ | $97.31 \pm 0.13$ | $83.71 \pm 0.60$ | 88.69 |
| $\alpha = 0.8$ | $89.47 \pm 0.61$ | $84.39 \pm 0.18$ | $97.41 \pm 0.15$ | $83.68 \pm 0.15$ | 88.74 |
| $\alpha = 0.9$ | $89.66 \pm 0.23$ | $85.04 \pm 0.28$ | $\mathbf{97.60 \pm 0.13}$ | $84.27 \pm 0.40$ | 89.14 |
| $\alpha = 1.0$ | $\mathbf{89.80 \pm 0.34}$ | $\mathbf{85.16 \pm 0.65}$ | $97.59 \pm 0.27$ | $\mathbf{84.67 \pm 0.82}$ | **89.30** |

(b) OfficeHome

|  | Art | Clipart | Product | Real | Average |
|---|---|---|---|---|---|
| $\alpha = 0.0$ | $65.99 \pm 0.17$ | $57.72 \pm 0.16$ | $78.36 \pm 0.08$ | $80.22 \pm 0.09$ | 70.57 |
| $\alpha = 0.1$ | $67.04 \pm 0.15$ | $58.09 \pm 0.11$ | $78.72 \pm 0.15$ | $80.36 \pm 0.08$ | 71.05 |
| $\alpha = 0.2$ | $\mathbf{67.46 \pm 0.28}$ | $58.45 \pm 0.13$ | $\mathbf{79.27 \pm 0.07}$ | $\mathbf{80.70 \pm 0.11}$ | **71.47** |
| $\alpha = 0.3$ | $66.98 \pm 0.24$ | $\mathbf{58.50 \pm 0.22}$ | $78.87 \pm 0.18$ | $80.45 \pm 0.06$ | 71.20 |
| $\alpha = 0.4$ | $67.04 \pm 0.32$ | $58.37 \pm 0.26$ | $78.57 \pm 0.09$ | $80.43 \pm 0.09$ | 71.10 |
| $\alpha = 0.6$ | $66.45 \pm 0.33$ | $57.99 \pm 0.20$ | $78.53 \pm 0.21$ | $80.22 \pm 0.11$ | 70.80 |
| $\alpha = 0.8$ | $66.56 \pm 0.23$ | $57.89 \pm 0.18$ | $78.50 \pm 0.12$ | $80.21 \pm 0.18$ | 70.79 |
| $\alpha = 1.0$ | $66.47 \pm 0.30$ | $57.96 \pm 0.09$ | $78.46 \pm 0.30$ | $79.97 \pm 0.29$ | 70.72 |

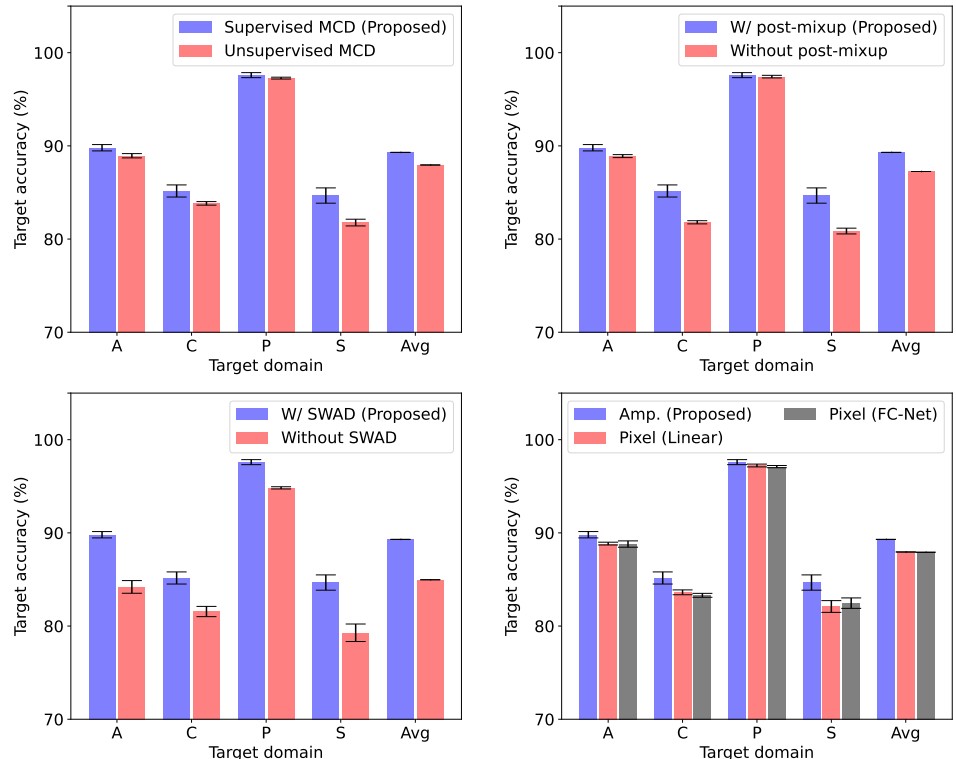

Figure 5: Ablation study of four different modeling choices: SMCD, post-mixup, SWAD, and amplitude generation (instead of pixel-based target image generation).

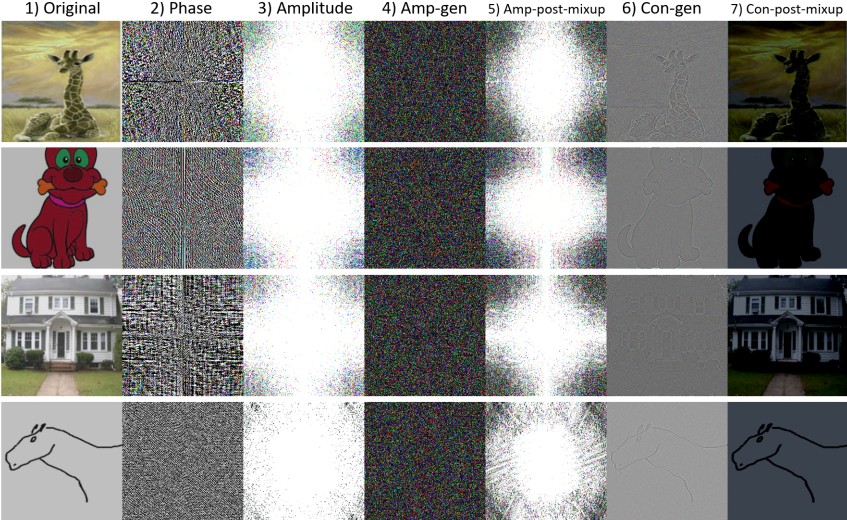

Figure 6: Visualisation of the generated amplitude and constructed images. The columns are (from left to right): 1) original image, 2) phase and 3) amplitude spectra after Fourier transform, 4) generated amplitude image, 5) post-mixup of 3 and 4, 6) constructed image from phase in 2) and generated amplitude image in 4) (by inverse Fourier transform), and 7) constructed image from phase 2 and the post-mixup amplitude 5.

on the PACS benchmark are shown in Table 13 for (a) ResNet-18 and (b) ResNet-50 backbones. The results indicate that improvement of the proposed AGFA over the existing DG methods is even more pronounced: averaged accuracies higher than the best prior method EFDMIX (Zhang et al., 2022) by about 10% for ResNet-18 and by about 7% for ResNet-50.

Table 10: Average accuracies on PACS with ResNet-18 backbone. Results on ERM, Mixup (Zhang et al., 2018), MixStyle (Zhou et al., 2021b), and EFDMix (Zhang et al., 2022) are excerpted from (Zhang et al., 2022).

| Algorithm | Art | Cartoon | Painting | Sketch | Avg |
|---|---|---|---|---|---|
| ERM | $77.0 \pm 0.6$ | $75.9 \pm 0.6$ | $96.0 \pm 0.1$ | $69.2 \pm 0.6$ | 79.5 |
| Mixup | $76.8 \pm 0.7$ | $74.9 \pm 0.7$ | $95.8 \pm 0.3$ | $66.6 \pm 0.7$ | 78.5 |
| MixStyle | $83.1 \pm 0.8$ | $78.6 \pm 0.9$ | $95.9 \pm 0.4$ | $74.2 \pm 2.7$ | 82.9 |
| EFDMix | $83.9 \pm 0.4$ | $\mathbf{79.4 \pm 0.7}$ | $\mathbf{96.8 \pm 0.4}$ | $75.0 \pm 0.7$ | 83.9 |
| (Proposed) AGFA | $\mathbf{84.5 \pm 0.6}$ | $78.5 \pm 0.5$ | $95.7 \pm 0.1$ | $\mathbf{80.9 \pm 0.2}$ | $\mathbf{84.9}$ |

Table 11: Average accuracies on Colored-MNIST with the four-layer ConvNet backbone. Results on competing methods are excerpted from (Gulrajani & Lopez-Paz, 2021).

| Algorithm | 0.1 | 0.2 | 0.9 | Avg |
|---|---|---|---|---|
| ERM | $72.7 \pm 0.2$ | $73.2 \pm 0.3$ | $10.0 \pm 0.0$ | 52.0 |
| IRM | $72.0 \pm 0.2$ | $73.2 \pm 0.0$ | $10.1 \pm 0.2$ | 51.8 |
| DRO | $72.7 \pm 0.3$ | $73.1 \pm 0.3$ | $10.0 \pm 0.0$ | 51.9 |
| Mixup | $72.4 \pm 0.2$ | $73.3 \pm 0.3$ | $10.0 \pm 0.1$ | 51.9 |
| MLDG | $71.4 \pm 0.4$ | $73.3 \pm 0.0$ | $10.0 \pm 0.1$ | 51.6 |
| CORAL | $71.8 \pm 0.4$ | $73.3 \pm 0.2$ | $10.1 \pm 0.1$ | 51.7 |
| MMD | $72.1 \pm 0.2$ | $72.8 \pm 0.2$ | $10.5 \pm 0.2$ | 51.8 |
| ADA | $72.0 \pm 0.3$ | $72.4 \pm 0.5$ | $10.0 \pm 0.2$ | 51.5 |
| CondADA | $72.2 \pm 0.3$ | $73.2 \pm 0.2$ | $10.4 \pm 0.3$ | 51.9 |
| (Proposed) AGFA | $72.6 \pm 0.1$ | $73.8 \pm 0.1$ | $10.5 \pm 0.1$ | 52.3 |

Table 12: Average accuracies on Rotated-MNIST with the four-layer ConvNet backbone. Results on competing methods are excerpted from (Gulrajani & Lopez-Paz, 2021).

| Algorithm | 0 | 15 | 30 | 45 | 60 | 75 | Avg |
|---|---|---|---|---|---|---|---|
| ERM | $95.6 \pm 0.1$ | $99.0 \pm 0.1$ | $98.9 \pm 0.0$ | $99.1 \pm 0.1$ | $99.0 \pm 0.0$ | $96.7 \pm 0.2$ | 98.1 |
| IRM | $95.9 \pm 0.2$ | $98.9 \pm 0.0$ | $99.0 \pm 0.0$ | $98.8 \pm 0.1$ | $98.9 \pm 0.1$ | $95.5 \pm 0.3$ | 97.8 |
| DRO | $95.9 \pm 0.1$ | $98.9 \pm 0.0$ | $99.0 \pm 0.1$ | $99.0 \pm 0.0$ | $99.0 \pm 0.0$ | $96.9 \pm 0.1$ | 98.1 |
| Mixup | $96.1 \pm 0.2$ | $99.1 \pm 0.0$ | $98.9 \pm 0.0$ | $99.0 \pm 0.0$ | $99.0 \pm 0.1$ | $96.6 \pm 0.1$ | 98.1 |
| MLDG | $95.9 \pm 0.2$ | $98.9 \pm 0.1$ | $99.0 \pm 0.0$ | $99.1 \pm 0.0$ | $99.0 \pm 0.0$ | $96.0 \pm 0.2$ | 98.0 |
| CORAL | $95.7 \pm 0.2$ | $99.0 \pm 0.0$ | $99.1 \pm 0.1$ | $99.1 \pm 0.0$ | $99.0 \pm 0.0$ | $96.7 \pm 0.2$ | 98.1 |
| MMD | $96.6 \pm 0.1$ | $98.9 \pm 0.0$ | $98.9 \pm 0.1$ | $99.1 \pm 0.1$ | $99.0 \pm 0.0$ | $96.2 \pm 0.1$ | 98.1 |
| DANN | $95.6 \pm 0.3$ | $98.9 \pm 0.0$ | $98.9 \pm 0.0$ | $99.0 \pm 0.1$ | $98.9 \pm 0.0$ | $95.9 \pm 0.5$ | 97.9 |
| C-DANN | $96.0 \pm 0.5$ | $98.8 \pm 0.0$ | $99.0 \pm 0.1$ | $99.1 \pm 0.0$ | $98.9 \pm 0.1$ | $96.5 \pm 0.3$ | 98.0 |
| (Proposed) AGFA | $98.1 \pm 0.1$ | $98.9 \pm 0.0$ | $99.0 \pm 0.0$ | $98.8 \pm 0.0$ | $99.0 \pm 0.0$ | $96.4 \pm 0.1$ | 98.0 |

### A.4.4 COMPARISON WITH PIXEL-BASED TARGET IMAGE GENERATION

Our Fourier-based target image generation is effective for preserving semantic class information from the source domains, thanks to the phase/amplitude separation. To see if non-Fourier-based generation also has similar property, we visualise adversarial target images generated by a purely pixel-based manner without phase/amplitude separation. For the linear pixel-based generator model (from 100-dim input noise to full image pixels), which performed slightly better than nonlinear ones in test accuracy, we show some examples in Fig. 7. Whereas the pixel-based generation is visually uninformative and looks like pure random noise, our Fourier-based generation contains salient object edge information that is closely related to class semantics.

Table 13: Single source domain generalisation results on PACS with (a) ResNet-18 and (b) ResNet-50 backbones. Each column shows test accuracies averaged over the rest three target domains. Results on ERM, MixStyle (Zhou et al., 2021b), and EFDMix (Zhang et al., 2022) are excerpted from (Zhang et al., 2022).

| (a) ResNet-18 | | | | | |
|---|---|---|---|---|---|
| Algorithm | Art | Cartoon | Painting | Sketch | Avg |
| ERM | $58.6 \pm 2.4$ | $66.4 \pm 0.7$ | $34.0 \pm 1.8$ | $27.5 \pm 4.3$ | 46.6 |
| MixStyle | $61.9 \pm 2.2$ | $71.5 \pm 0.8$ | $41.2 \pm 1.8$ | $32.2 \pm 4.1$ | 51.7 |
| EFDMix | $63.2 \pm 2.3$ | $73.9 \pm 0.7$ | $42.5 \pm 1.8$ | $38.1 \pm 3.7$ | 54.4 |
| (Proposed) AGFA | $\mathbf{74.2 \pm 1.1}$ | $\mathbf{77.5 \pm 0.6}$ | $\mathbf{48.5 \pm 2.6}$ | $\mathbf{58.3 \pm 0.9}$ | **64.6** |
| (b) ResNet-50 | | | | | |
| Algorithm | Art | Cartoon | Painting | Sketch | Avg |
| ERM | $63.5 \pm 1.3$ | $69.2 \pm 1.6$ | $38.0 \pm 0.9$ | $31.4 \pm 1.5$ | 50.5 |
| MixStyle | $73.2 \pm 1.1$ | $74.8 \pm 1.1$ | $46.0 \pm 2.0$ | $40.6 \pm 2.0$ | 58.6 |
| EFDMix | $75.3 \pm 0.9$ | $77.4 \pm 0.8$ | $48.0 \pm 0.9$ | $44.2 \pm 2.4$ | 61.2 |
| (Proposed) AGFA | $\mathbf{79.8 \pm 0.9}$ | $\mathbf{81.7 \pm 0.6}$ | $\mathbf{48.6 \pm 0.5}$ | $\mathbf{64.6 \pm 1.1}$ | **68.7** |

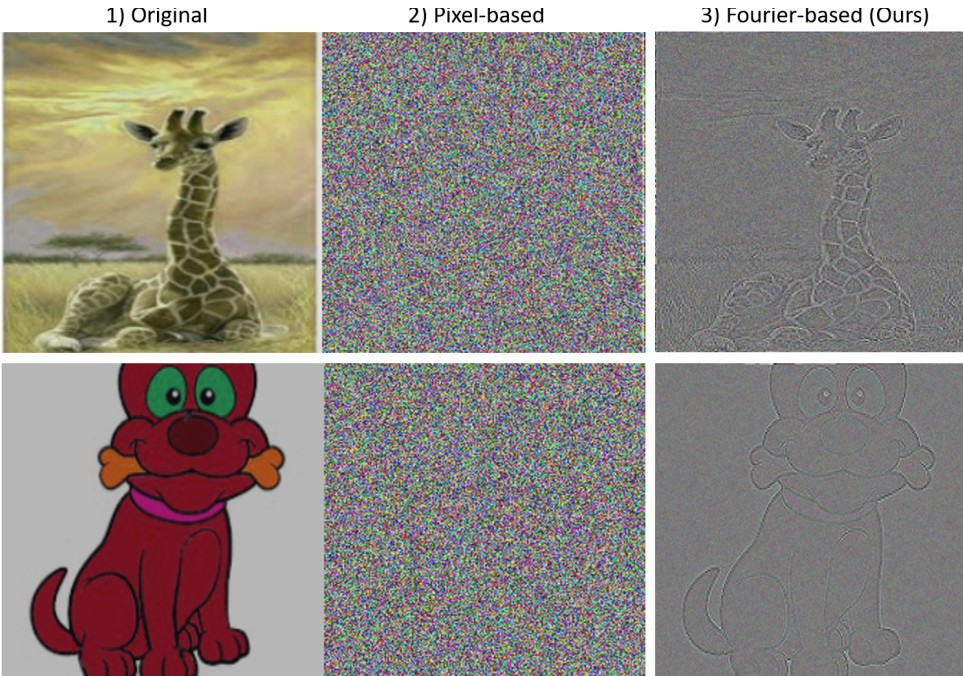

Figure 7: Comparison between pixel-based and our Fourier-based generated target images. Whereas the pixel-based generation is visually uninformative and looks like pure random noise, our Fourier-based generation contains salient object edge information that is closely related to class semantics.

