# OpenReview forum: "Domain Generalisation via Domain Adaptation: An Adversarial Fourier Amplitude Approach"
_ICLR.cc/2023/Conference — ICLR 2023 poster_

### Official Review · Reviewer_pXLL · 2022-10-24

**Confidence:** 3
**Correctness:** 3
**Technical Novelty And Significance:** 2
**Empirical Novelty And Significance:** 3
**Recommendation:** 5

**Clarity, Quality, Novelty And Reproducibility:**

Clarity: the paper is well written and clear
Quality: the experiments seem extensive to me, however the poor visual quality of the target domains makes me question whether there is more to this story.
Novelty: Novelty is limited, to replacing amplitude mixup by adversarial training and the supervised MCD loss.
Reproducibly: The method is not too complicated, perhaps it can be reproduced but no code was released.

**Strength And Weaknesses:**

Strengths:
* The paper is clearly written and a pleasure to read.
* The main ideas are intuitive
* Augmenting MCD with the true labels is probably novel (as is an aspect of this setting)
* Results indeed outperform many previous methods

Weakness
* Synthesizing domains in fourier space clearly does not generate realistic images - and so the idea diesn't really work in that sense - even though it did generate improvements on the baseline. The method resolves some of the issue by using mixup, but this is obviously not the main idea the paper was selling. It makes the reviewer ponder if this is the fourier parametrization is the correct one for synthesising the target domain.

* The method is only slightly novel compared to the previous baseline of fourier domain mixup. Most of the advantage comes form SWAD, which is orthogonal.

**Summary Of The Paper:**

The paper tackles domain generalisation by synthesizing the hardest domain, and then using domain adaptation techniques. Domains are parametrized in fourier space using the amplitudes which the phases are taken from the original images. The hardest target domains synthesized by adversarial training. The classifier is learned via MCD, which is modified here as the target labels are known. Results seem better than mixup in fourier space.

**Summary Of The Review:**

This is a well written paper, which tells a nice story and presents strong results. The novelty is a little limited and the domain synthesis is raises some doubts. I am borderline on this paper and waiting for the rebuttal and other reviews.

---

> ### Author Response · Authors · 2022-11-18
> **Responses to Reviewer pXLL**
>
> **1. Synthesizing domains in Fourier space clearly does not generate realistic images. The method resolves some of the issue by using mixup, but this is obviously not the main idea the paper was selling. It makes the reviewer ponder if this is the Fourier parametrization is the correct one for synthesising the target domain. The poor visual quality of the target domains makes me question whether there is more to this story.**
>
> We address this point from three perspectives as follows:
>
> First, we emphasise that the synthesised target domain images do not necessarily have to be visually natural looking in order to be useful for DG. We generate worst-case images based on training (source domain) data. Since these are worst-case images it is not surprising that they might look un-natural -- any natural image must be an easier-than-worst-case example by definition. However, that does not prevent our synthetic worst-case images from being effective in making the learned classifier more robust to unseen changes in test domains, as proved in our empirical results.
>
> Second, we remark that from a first prinicples point of view, our Fourier-domain parametrisation is a reasonable strategy mainly to help meet the three criteria for DG (Pg 4, below Eq7), especially the one that the target domain has to lie in the same semantic class space as the source domains (C2).
>
> Finally, we remark that our Fourier-based approach defines a particular manifold on which to search for the worst case image. As an alternative, we consider what if we abandoned this manifold and search for the worst case image in pure pixel space? To this end we visualise adversarial target images generated by a purely pixel-based manner without phase/amplitude separation. We include some examples in Fig. 7 in Appendix A.4.4 of the revised paper. It shows that the pixel-based generated target images look like pure random noise. Please compare the pixel-space worst-case images with the Fourier-domain worst-case images in Fig. 7. We can see that for the Fourier-domain synthesized example is sufficiently perceptible in edge form that a human can recognise the object. Meanwhile the pixel-space synthesised example is completely unrecognisable for a human reader. This comparison shows that while Fourier-space may not be the best possible solution for image synthesis, it is clearly a better than the obvious pixel space alternative. Together with the fact that our ablation study (Table 2), shows that pixel-based (non-Fourier) synthesis leads to inferior quantitative performance, this confirms the value of our contribution of doing worst-case image synthesis, and doing so in Fourier domain.
>
>
> **2. The method is only slightly novel compared to the previous baseline of Fourier domain mixup. Most of the advantage comes form SWAD, which is orthogonal.**
>
> The main novelty of the paper is in two folds: 1) adversarially synthesizing the Fourier amplitude images, which is different from the simple amplitude mixup in the previous work, and 2) recasting the domain generalisation as a domain adaptation problem by the worst-case target domain synthesis. The latter contribution especially is completely novel in methodology and scope, with nothing like this having been considered by the Fourier domain mixup paper. Empirically our approach is clearly better than SWAD alone and also better than SWAD + Fourier domain mixup. This demonstrates our empirical contribution is significant compared to both SWAD and Fourier domain mixup.

---

### Official Review · Reviewer_71uu · 2022-10-24

**Confidence:** 3
**Correctness:** 3
**Technical Novelty And Significance:** 3
**Empirical Novelty And Significance:** 3
**Recommendation:** 5

**Clarity, Quality, Novelty And Reproducibility:**

Clarity: Good. This paper is well-written, it is enjoyable to read.

Quality: Good. Experiments are comprehensive and show the effectiveness of the proposed method.

Novelty: A little marginal. MCD is an existing approach. And why choosing Bayesian hypothesis modeling is not clear.


**Strength And Weaknesses:**

Strengths:
+ This paper studies an important problem
+ This paper is well-written, it is enjoyable to read.
+ Experiments are comprehensive and show the effectiveness of the proposed method.

Weaknesses:
+ The contributions are a little limited considering that MCD is an existing approach.
+ This paper should concentrate more on true solutions in DG. More discussions about the true solution in DG should be provided.
+ Why choosing Bayesian hypothesis modeling? It should be clearly discussed.
+ More insightful discussions should be provided. For example, how to improve the generalization ability of deep models using other adversarial DA approaches (e.g. DANN and MDD)?
[1] Domain-adversarial training of neural networks
[2] Bridging theory and algorithm for domain adaptation.

**Summary Of The Paper:**

This paper targets at solving domain generalization. The authors tackle the DG problem as DA task where the worst-case `target' domain is adversarially synthesized. Specifically, they generate Fourier amplitude images and combine them with source domain phase samples. MCD is further exploited to relate the target domain performance to the discrepancy of classifiers in the model hypothesis space. Bayesian hypothesis modeling is adopted to make adversarial MCD minimization feasible. Experiments show the effectiveness of the proposed method.

**Summary Of The Review:**

Strengths:
+ This paper studies an important problem
+ This paper is well-written, it is enjoyable to read.
+ Experiments are comprehensive and show the effectiveness of the proposed method.

Weaknesses:
+ The contributions are a little limited considering that MCD is an existing approach.
+ This paper should concentrate more on true solutions in DG. More discussions about the true solution in DG should be provided.
+ Why choosing Bayesian hypothesis modeling? It should be clearly discussed.
+ More insightful discussions should be provided. For example, how to improve the generalization ability of deep models using other adversarial DA approaches (e.g. DANN and MDD)?
[1] Domain-adversarial training of neural networks
[2] Bridging theory and algorithm for domain adaptation.

---

> ### Author Response · Authors · 2022-11-18
> **Request for clarification on Reviewer(71uu)'s question**
>
> Regarding the reviewer's question,
>
> **``This paper should concentrate more on true solutions in DG. More discussions about the true solution in DG should be provided''**,
>
> we do not understand what is meant by **``true solutions in DG''**. Could you elaborate more so that we can address the question properly?

---

> > ### Comment · Reviewer_71uu · 2022-11-23
> > **Clarification**
> >
> > I would like to see a discussion about what is the true solution to achieve DG. DG aims to achieve OOD generalization by using only source data. However, in my opinion, it is more important to improve the generalization ability in DG (DA is transductive learning while DG is inductive learning). As a result,  in DG, DA-based approaches highly rely on the quality of the source domain and the relationship between the source and the unseen target. Although such a solution is proposed under the current conditions, I do not think it can be treated as a true solution in DG. Pessimistically, I think it may mislead the development of DG. It is not clear to me whether the proposed approach can synthesize the unseen target domain. I would like to know the opinion of the authors.

---

> > > ### Author Response · Authors · 2022-11-25
> > > **Responses to the clarified question**
> > >
> > > Thank you. Our understanding about the reviewer's question is how the inductive DG problem can be tackled in principle by a method that relies on the (MCD) theorem developed for transductive DA. (Please correct us if our understanding is wrong.) Our answer to the question is yes we can. It is correct that DA is transductive and DG is inductive. But since we do consider **all** plausible target domains during DA training (via our worst-case target synthesis), the approach becomes inductive by definition. Here, by plausible we mean the class semantics of the source domains is preserved in the target domains.
> > >
> > > Another concern is whether the proposed method can synthesise the unseen target domain or not. It may be difficult to judge this visually since our Fourier-based approach defines a particular manifold on which to search for the worst-case target domain images. However, the superiority of our approach in empirical comparison to various existing mixup-based strategies to generate target domains and the baseline ERM, clearly confirms the value of the proposed worst-case target synthesis in improving DG prediction performance. We also note that our Fourier-space target generation may not be the best possible solution for image synthesis, but it is clearly better than the obvious pixel space alternative as we demonstrate in Fig. 7 in Appendix A.4.4 of the revised paper.

---

> ### Author Response · Authors · 2022-11-18
> **Responses to Reviewer 71uu**
>
> **1. The contributions are a little limited considering that MCD is an existing approach.**
>
> The MCD is a theorem developed for Domain Adaptation with the assumption that the target domain inputs are known at the training stage. How to apply MCD to Domain Generalisation is not obvious since the target domain data are not available. Our main contribution is to extend the MCD to DG by forming a mini-max problem with the worst-case target domain generation via adversarial Fourier amplitude synthesis. Since MCD previously considered fixed and known domains, then far from being a minor extension of MCD, this is a dramatic extension of both algorithm and scope.
>
>
> **2. Why choosing Bayesian hypothesis modeling? It should be clearly discussed.**
>
> As stated in the first paragraph of Sec. 3, we need to represent and optimize the hypothesis space for minimizing the MCD loss, and the Bayesian modeling is very adequate for that purpose: A hypothesis space can be represented as a distribution over classifiers.
>
>
> **3.  How to improve the generalization ability of deep models using other adversarial DA approaches (e.g. DANN and MDD)?**
>
> Although we have not tried other adversarial approaches as the reviewer pointed, we think that it is potentially possible to apply them in place of the Fourier amplitude synthesis or MCD. This can be possible future work.

---

### Official Review · Reviewer_5QDe · 2022-10-24

**Confidence:** 5
**Correctness:** 4
**Technical Novelty And Significance:** 3
**Empirical Novelty And Significance:** Not applicable
**Recommendation:** 6

**Clarity, Quality, Novelty And Reproducibility:**

The clarity, quality, and novelty are pretty good. Not sure if the paper results can be reproduced.

**Strength And Weaknesses:**

Strength
(1)	The proposed method is novel and well-motived.
(2)	This paper gives a new perspective on DG by finding the worst-case ib an adversarial way and posing it as a domain adaptation (DA) task.
(3)	Empirical results indicate that the proposed method greatly outperforms the other methods.
Weakness:
(1)	Even the generated target domain is the "worst case ", it is not as same as the true target domains. It would be better to discuss this in theoretical analysis.
(2)	Why do not directly minimize e^*(\mathcal{H} ; S, T) by one supervisor loss if you have known the label of generated target domain.
(3)	The results of CMNIST and RMNIST in the DomainBad benchmark are not provided in this paper.
(4)	In this paper, there is no comparison with other methods [3,4] in different backbone networks, such as resnet18 in PACS.
(5)	Only the result of the leave-one-domain-out setting is provided. Can you give the result in single source generalization setting as EFDMix[4]?
[1] Minyoung Kim, Pritish Sahu, Behnam Gholami, and Vladimir Pavlovic. Unsupervised Visual Domain Adaptation: A Deep Max-Margin Gaussian Process Approach. CVPR2019.
[2] Zhihe Lu, Yongxin Yang, Xiatian Zhu, Cong Liu, Yi-Zhe Song, and Tao Xiang. Stochastic classifiers for unsupervised domain adaptation. CVPR2020.
[3] Kaiyang Zhou, Yongxin Yang, Yu Qiao, and Tao Xiang. Domain generalization with mixstyle. ICLR2021.
[4] Yabin Zhang, Minghan Li, Ruihuang Li, Kui Jia and Lei Zhang. Exact Feature Distribution Matching for Arbitrary Style Transfer and Domain Generalization. CVPR2022.


**Summary Of The Paper:**

This paper introduces a new domain generalization (DG) method by synthesizing the virtual target domain samples in Fourier domain and exploiting the maximum classifier discrepancy (MCD) principle in source domain and generated target domain. It also gives a modified MCD loss, which is suitable for DG task. The experiments show the effectiveness of the proposed method.

**Summary Of The Review:**

The paper proposes a novel and well-motived method for domain generalization. However, there are lacks of some experiments and more discussion about the theory and the relation between SMCD and other losses. If you solve the problems, I am willing to improve my rating.

---

> ### Author Response · Authors · 2022-11-18
> **Responses to Reviewer 5QDe**
>
> **1. Even the generated target domain is the ``worst case'', it is not as same as the true target domains. It would be better to discuss this in theoretical analysis.**
>
> What are the true target domains? In DG, we do not know these in advance, except that the target domain has the same semantic class space as that of the source domain. Given that the true target domain is unknowable in a DG context, we believe that preparation for the worst-case target domain is a reasonable strategy to take.
>
>
> **2. Why do not directly minimize** $e^*(\mathcal{H} ; S, T)$ **by one supervisor loss if you have known the label of generated target domain.**
>
> Our modified MCD loss (dubbed SMCD) does exactly what the reviewer questioned. We minimize $e^*(\mathcal{H} ; S, T)$ with the max margin principle.
>
>
> **3. The results of CMNIST and RMNIST in the DomainBad benchmark are not provided in this paper.**
>
> Although relatively smaller and easier datasets than the other DomainBed tasks that we already present, we now also test our method on the Coloured-MNIST and Rotated-MNIST datasets as the reviewer suggested. Following the experimental protocols including the four-layer ConvNet backbone as in (Gulrajani & Lopez-Paz, 2021), the test accuracies are reported in Table 11 (Colored-MNIST) and Table 12 (Rotated-MNIST) in Appendix A.4.2 of the revised paper. As shown, all approaches including ours perform equally well on these datasets.
>
>
> **4. In this paper, there is no comparison with other methods [3,4] in different backbone networks, such as resnet18 in PACS. [3] Kaiyang Zhou, Yongxin Yang, Yu Qiao, and Tao Xiang. Domain generalization with mixstyle. ICLR2021. [4] Yabin Zhang, Minghan Li, Ruihuang Li, Kui Jia and Lei Zhang. Exact Feature Distribution Matching for Arbitrary Style Transfer and Domain Generalization. CVPR2022.**
>
> We take reviewer's suggestion, and have conducted experiments with the ResNet-18 backbone on the PACS dataset. The results are summarised in Table 10 in  Appendix A.4.1 of the revised paper. Compared to the recent approaches MixStyle (ICLR'21) and EFDMix (CVPR'22), our approach AGFA again shows higher performance. For instance, the average test accuracies are: MixStyle $=82.9$, EFDMix $=83.9$, Ours $=84.9$.
>
>
> **5. Only the result of the leave-one-domain-out setting is provided. Can you give the result in single source generalization setting as EFDMix[4]?**
>
> Thanks for this suggestion. We agree with the reviewer that the single source generalisation setting is another reasonable experimental setting. We performed the experiments,  and the results on the PACS benchmark are shown in Table 13 of the revised paper for (a) ResNet-18 and (b) ResNet-50 backbones. The results indicate that improvement of the proposed AGFA over the existing DG methods is even more pronounced: We outperform the previous best method  EFDMIX (Zhang et al. CVPR'22) by about $10\%$ for ResNet-18 and by about $7\%$ for ResNet-50.

---

### Official Review · Reviewer_M2ax · 2022-10-26

**Confidence:** 3
**Correctness:** 3
**Technical Novelty And Significance:** 2
**Empirical Novelty And Significance:** 3
**Recommendation:** 6

**Clarity, Quality, Novelty And Reproducibility:**

The paper is well articulated and provides the extra context in the appendix.
The empirical part is good and the method is decent and relatively simple, albeit working surprisingly well across several datasets.
I would say that pseudocode is just enough to reproduce it, but that assumes that the hyperparameters described in the main text are what one needs to reproduce the results rather than trying to magically grid search parameters. In addition, it would be nice to have a response from the authors as to whether they plan to release the full code.

****Authors please be consistent in the use of American vs British English - choose whichever you prefer but stick with it across the paper.****

**Strength And Weaknesses:**

Strengths:

a) generating target images for modelling worst-case scenario
b) repurposing mean classifier discrepancy to propose a supervised variant that leverages the labelled generated target domain data
c) competitive performance
d) relative simple implementation

Weakness:

a) not a weakness per se, but I am trying to get my head round why fourier amplitude for adversarially generating images.... is that to create bad but not too bad images deliberately to force a better generalisation performance? It sounds rather arbitrary to me (not in a negative way) so I would like to know what the thought process was to arrive to that approach vs trying other ways,

**Summary Of The Paper:**

This paper proposes an approach to domain generalisation whereby they use Fourier to generate synthetic images which are then used as a worst-case target domain to improve the model's robustness and therefore its generalisation. This is reminiscent of a domain adaptation approach. The amplitude-based generator that creates the synthetic images is trained in an adversarial manner and is combined with the phase source domain data. Surprisingly, the method performs competitively compared to other methods proposed.


**Summary Of The Review:**

This is a good submission that seems to perform well. Authors have run several experiments some appearing in the main text and some others in the appendix. There is a good empirical component and the novelty is fair. I think this is a decent submission to be considered for acceptance.

---

> ### Author Response · Authors · 2022-11-18
> **Responses to Reviewer M2ax**
>
> **1. Why Fourier amplitude for adversarially generating images.... is that to create bad but not too bad images deliberately to force a better generalisation performance? It sounds rather arbitrary to me (not in a negative way) so I would like to know what the thought process was to arrive to that approach vs trying other ways.**
>
> Yes, this is the motivation of why we considered the Fourier amplitude synthesis. As stated in the paper (below Eq7 on Pg4), we need to meet the three criteria in DG, and the Fourier amplitude synthesis has a key role in generating images of novel styles but under the same semantic class  as the source domains.
>
>
> **2. It would be nice to have a response from the authors as to whether they plan to release the full code.**
>
> Yes. We will publicize the codes to reproduce the results in the paper, upon acceptance.
>
>
> **3. Authors please be consistent in the use of American vs British English - choose whichever you prefer but stick with it across the paper.**
>
> Thank you for pointing this out. We have refined it in our revised paper.

---

### Author Response · Authors · 2022-11-18
**Revised manuscript with clarifications and additional experiments**

We thank all reviewers for their insightful and constructive comments/questions. Our responses to reviewers' individual comments/questions are placed after each review.

---

### Decision · Program_Chairs · 2023-01-20

**Decision:**

Accept: poster

**Justification For Why Not Higher Score:**

This paper is evaluated having contrasting comments, which have been fixed but still the reviewers were on the fence, so I think poster presentation is more suitable than higher scores.

**Justification For Why Not Lower Score:**

The paper was accepted since, despite the initial criticisms, authors replied properly to all comments, and in these conditions I don't see reasons to reject it

**Metareview: Summary, Strengths And Weaknesses:**

This paper received contrasting reviews before and after authors' rebuttal, ratings are slightly above and below the threshold (6, 6, 5, 5), even if one of the positive reviewer seems not against rejection.
All reviewers seem to have arguments in favor and against this work.
Main issues regard the insufficient justification of the use of the Fourier domain, weak novelty (just for some reviewers), and missing experimental results. About the use of the Fourier domain there are also other comments, e.g., regarding the fact that the method does not generate realistic images, but this is not a big issue to my opinion.
Overall, all the raised comments seem to be well addressed by the authors in their rebuttal: the paper results sufficiently novel, the methodological contribution appears significant, the experimental analysis is comprehensive, also considering all the requests done by the reviewers, hence, my evaluation is that this paper is acceptable for publication to ICLR 23.


**Note From Pc:**

if the above contains the word "oral" or "spotlight" please see: "oral" presentation means -> notable-top-5% and "spotlight" means -> notable-top-25%. As stated in our emails, we are disassociating presentation type from AC recommendations

**Summary Of Ac-Reviewer Meeting:**

Reviewer pXLL (5) was slightly negative and posted the same issues raised in his review, especially the justification about the involvement of the Fourier domain and novelty, but in the end, he can also agree on acceptance. The comment about the involvement of the Fourier domain is not so consistent to my opinion, its use should not be so largely justified to my opinion.
After reviewer pXLL's comments, reviewer M2ax (scoring 6) agrees with them and seems to turn to slightly negative even if he did not lower his score. Overall, my impression is that none is safe with his evaluation, staying in the middle.
The other reviewers did not react to my solicitations, but had discussion with authors in additional rounds of review.